# MAP: Multi-Human-Value Alignment Palette

**Xinran Wang**[1]  **Qi Le**[1]  **Ammar Ahmed**[1]  **Enmao Diao**  **Yi Zhou**[2]  **Nathalie Baracaldo**[2]
**Jie Ding**[1]  **Ali Anwar**[1]
[1]University of Minnesota      [2]IBM Research
{wang8740, le000288, ahme0599, dingj, aanwar}@umn.edu,
diao_em@hotmail.com, yi.zhou@ibm.com, baracald@us.ibm.com

## Abstract

Ensuring that generative AI systems align with human values is essential but challenging, especially when considering multiple human values and their potential trade-offs. Since human values can be personalized and dynamically change over time, the desirable levels of value alignment vary across different ethnic groups, industry sectors, and user cohorts. Within existing frameworks, it is hard to define human values and align AI systems accordingly across different directions simultaneously, such as harmlessness, helpfulness, and positiveness. To address this, we develop a novel, first-principle approach called Multi-Human-Value Alignment Palette (MAP), which navigates the alignment across multiple human values in a structured and reliable way. MAP formulates the alignment problem as an optimization task with user-defined constraints, which define human value targets. It can be efficiently solved via a primal-dual approach, which determines whether a user-defined alignment target is achievable and how to achieve it. We conduct a detailed theoretical analysis of MAP by quantifying the trade-offs between values, the sensitivity to constraints, the fundamental connection between multi-value alignment and sequential alignment, and proving that linear weighted rewards are sufficient for multi-value alignment. Extensive experiments demonstrate MAP's ability to align multiple values in a principled manner while delivering strong empirical performance across various tasks. Our code is available at https://github.com/wang8740/MAP.

## 1 Introduction

Recent advancements in artificial intelligence (AI) have highlighted the critical need for aligning AI systems with human values, a concept known as *human value alignment* (Griffith et al., 2013; Arumugam et al., 2019; Gabriel, 2020). The alignment can serve the purpose of generating outcomes that are better suited for human ethics (Griffith et al., 2013), personalized needs (Kirk et al., 2024), or reduced harmful content (Bai et al., 2022). This alignment has traditionally been pursued by adjusting AI behavior to adhere to specific attributes via preference datasets or reward functions. This process involves finetuning the original model according to the optimization problem:

$$\max_{p \in \mathcal{P}} \mathbb{E}_{x \sim \mathcal{D}, y \sim p(\cdot \mid x)} \left\{ R(x, y) - \beta D_{\mathrm{KL}} \big( p(\cdot \mid x) \,\|\, p_0(\cdot \mid x) \big) \right\}. \tag{1}$$

Here, $\mathcal{P}$ denotes the class of all distributions, $p_0$ is the distribution that represents the generative model to align, $p$ is the distribution that represents the aligned model, $R$ is a reward function that quantifies the preference level of any given pair of prompt $x$ and generation $y$, $D_{\mathrm{KL}}$ measures the KL-divergence, and $\beta > 0$ is a regularization hyperparameter. This formulation has deep conceptual roots in the Bayesian decision theoretic framework (Bissiri et al., 2016). Specifically, if we consider $x$ as observed data and $y$ as a parameter $\theta$, the problem (1) can be expressed as $\mathbb{E}_{\theta \sim p(\cdot)} \big\{ \log p(x \mid \theta) - D_{\mathrm{KL}} \big( p(\cdot) \,\|\, p_0(\cdot) \big) \big\}$. This formulation yields the solution $p_x(\theta) \propto p_0(\theta) p(x \mid \theta)$, which is precisely Bayes' Rule. However, while this formulation provides an elegant interpretation of how AI models can be adjusted to reflect new information or preferences, it may not fully capture the complexity required when aligning AI systems to multiple, potentially conflicting human values. For example, a healthcare-related large language model must deliver information that is not only precise but also easy-to-understand and harmless (Gebreab et al., 2024). Similarly, a customer service

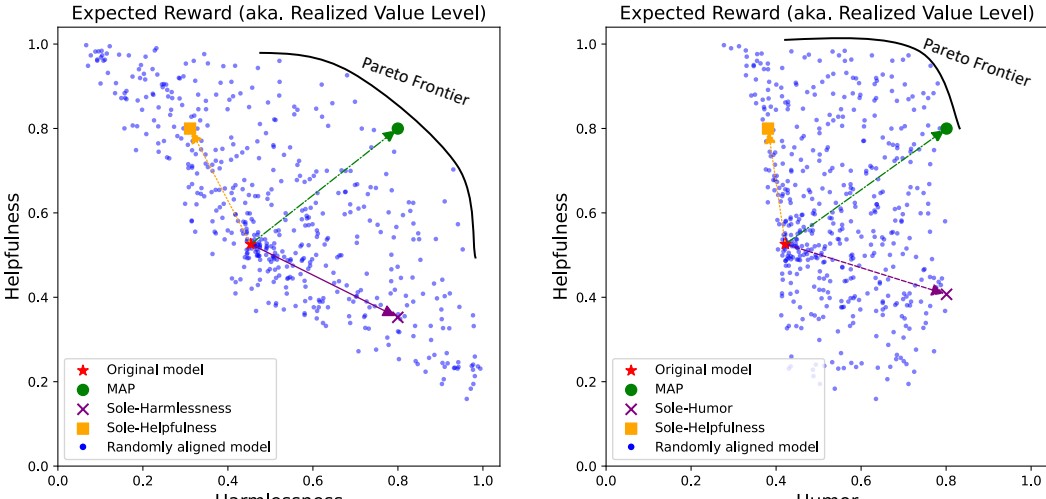

Figure 1: Expected reward (realized value level) of generated content having Harmlessness (left) and Humor (right) versus Helpfulness for various models aligned from the Llama2-7B-chat model (Touvron et al., 2023). Each blue dot represents the expected rewards $\mathbb{E}_{x \sim \mathcal{D}, y \sim p(\cdot|x)} r(x, y)$ with $r$ trained from Anthropic Harmless preference data ($r_{\text{Harmlessness}}$) (Yang et al., 2024), Helpfulness preference data ($r_{\text{Helpfulness}}$) (Yang et al., 2024), and Humor classifier ($r_{\text{Humor}}$) (Dhiab, 2023). The expected rewards are numerically obtained by solving (1) with $R = \lambda_1 r_{\text{Harmlessness}} + \lambda_2 r_{\text{Helpfulness}} + \lambda_3 r_{\text{Humor}}$, where $\lambda_1, \lambda_2, \lambda_3 \geq 0$ are randomly generated, and quantile-transformed to the scale of 0 to 1. Arrows indicate the transition from the original model to aligned models, either using the proposed approach (MAP) or a single reward function.

chatbot optimized for efficiency may compromise on politeness or empathy (Kolasani, 2023). These scenarios highlight the need to balance diverse human values, which often requires more nuanced solutions than a single-dimensional alignment approach can offer.

**Related work.** Centered around the formulation (1), a standard approach is the reinforcement learning from human feedback (RLHF) (Griffith et al., 2013; Arumugam et al., 2019; Bai et al., 2022; Dai et al., 2024) that first trains a reward model $R$ based on pairwise comparison or ranking data to serve as a proxy for human preferences, and then uses reinforcement learning with the reward model to address the optimization problem (1). An alternative method known as direct preference optimization (DPO) (Rafailov et al., 2023) directly optimizes an empirical risk based on the Bradley-Terry loss (Bradley & Terry, 1952) applied to the preference data, using an implicit reward in the form of $r(x, y) \stackrel{\Delta}{=} \beta \log(p(y \mid x)/p_0(y \mid x))$. To address multiple human values, recent work on aligning foundation models has used multi-objective reinforcement learning (MORL) (Barrett & Narayanan, 2008; Li et al., 2020; Wu et al., 2024). Existing studies focus on approximating trade-offs among values utilizing a linear scalarization method that combines either reward functions or data sources, for both RLHF and DPO approaches (Bai et al., 2022; Rame et al., 2023; Dai et al., 2024). Alternatively, some studies utilize specific choices of weights to manage trade-offs (Dognin et al., 2024). A recent technique named "rewarded soup" was introduced to efficiently compute the aligned models under linear scalarization (Rame et al., 2023). It first separately fine-tunes multiple models, each with a particular reward function, and then aggregates these networks by linearly interpolating their weights. This method aims to approximate the ensemble of fine-tuned models that would otherwise result from optimizing a single reward composed of various linear combinations of individual reward functions, namely $R \stackrel{\Delta}{=} \sum_{i=1}^{m} \lambda_i r_i$ with random $\boldsymbol{\lambda} = [\lambda_1, \ldots, \lambda_m]$. A similar idea was applied to DPO (Zhou et al., 2024), where the DPO-aligned models under individual values are interpolated to approximate the Pareto Frontier. Further related work is provided in Appendix B.5.

**Challenges.** Aligning AI models to multiple values simultaneously presents several unresolved challenges. First, as demonstrated in Figure 1, aligning with one value, such as helpfulness, harmlessness, or humor, could inadvertently diminish another. This motivates a critical question: *How can we quantify and enhance multiple human values concurrently without compromise?*

Moreover, in the RLHF approach as described by the problem (1), it is unclear how to specify the hyperparameter $\beta$ and reward function $R$ so that the aligned model $p$ improves upon, or at least not

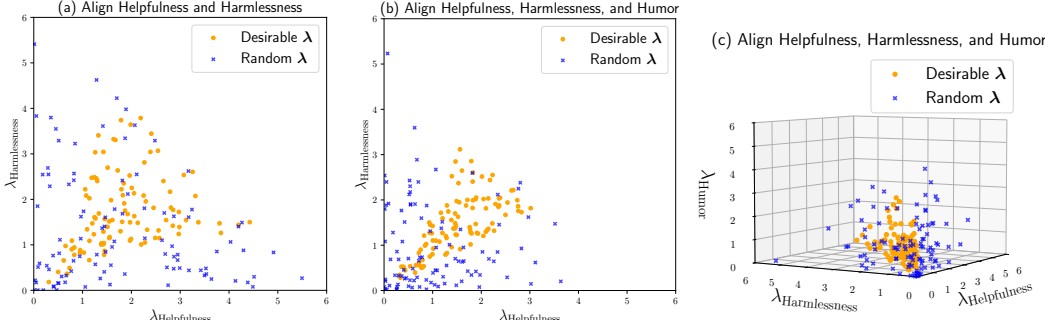

Figure 2: Randomly sampled $\boldsymbol{\lambda}$ that represent all the possible $\boldsymbol{\lambda}$ whose $\ell_1$-norm is less than 6 and its subset of all the desirable $\boldsymbol{\lambda}$ in aligning the OPT-1.3B model towards (a) two values: Helpfulness and Harmlessness, (b) three values: adding Humor, and (c) the same three values visualized in 3D. A desirable $\boldsymbol{\lambda}$ means it produces Pareto improvement over all the values. The sampling procedure for $\boldsymbol{\lambda}$ is the same as outlined in Section 3.3.

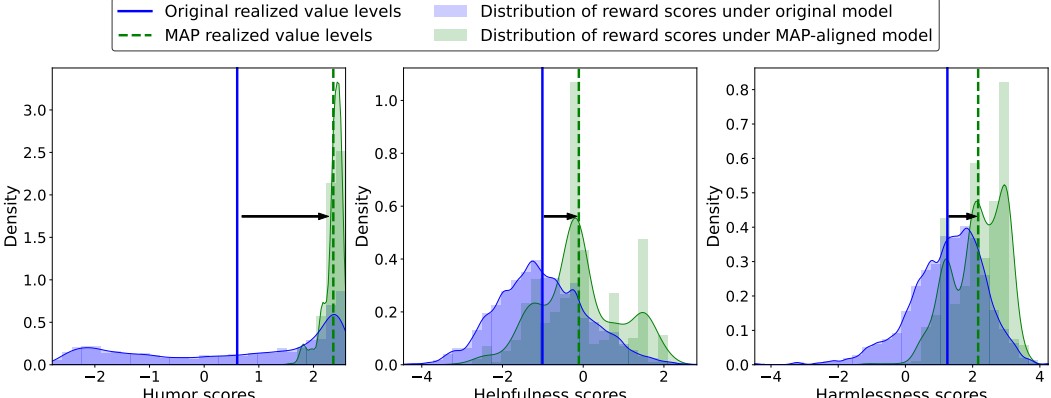

Figure 3: Distribution of reward scores before and after aligning the Llama2-7B-chat model towards three values: Humor (left plot), Helpfulness (middle plot), and Harmlessness (right plot), using the proposed MAP. This alignment involves a user-specified palette designed to shift the expected rewards, also referred to as realized value levels, toward the 80% quantile of the pre-alignment distributions.

worse than, $p_0$ in all human values in one shot (without trial-and-error). Even if we can try all possible combinations, there is no theoretical justification that linearly combining individual reward functions is sufficient to obtain the Pareto Frontier. The DPO method, while simplifying the alignment process through a direct empirical risk optimization, still does not address the issue of integrating multiple data sources, which have their underpinning still at the problem (1). Recent studies have demonstrated the sensitivity of results to different weights used in these aggregations (Bai et al., 2022). To highlight this point, in Figure 2, we visualize the range of possible $\boldsymbol{\lambda}$ and of desirable $\boldsymbol{\lambda}$ (which actually admits Pareto improvement on all the values). Figure 2(b) shows how adding additional value-to-align narrows the range of desirable $\boldsymbol{\lambda}$ compared with Figure 2(a).

**Contributions.** We introduce the Multi-Human-Value Alignment Palette (MAP), a principled approach to rigorously aligning multi-dimensional values with provable guarantees. Similar to an artist's color palette, MAP enables the blending of multiple human values to "paint" AI behavior with a broad spectrum of preference shades. In Figure 3, we illustrate how MAP allows users to precisely customize and control the level of improvement for all values in an interpretable manner. The proposed MAP introduces several technical innovations and contributions:

• Formulation. We propose a novel problem formulation that allows one to align multiple human values using user-defined constraints, which we term "value palettes." Each palette acts as a constraint that represents a preferred level of alignment, allowing us to "MAP" from any targeted value levels specified by the user to a particular reward function for (1). This precise one-to-one mapping ensures exact adjustments to model behavior.

• Theory. We provide theoretical analysis within the MAP framework quantifies the representation of the solution, its sensitivity to changes in the value palette, and its feasible operational range. This leads to a deeper understanding of the inherent trade-offs among various values. Furthermore,

we investigate the range of realizable value levels and demonstrate that a linear combination of individual reward functions, is sufficient to reach the Pareto Frontier. We also establish a crucial link between multi-value alignment and sequential alignment, showing that cyclically updating each value multiple times achieves equivalent results to a single execution of MAP.

• Computational methodology. Based on our theoretical derivations, we propose a computational method to ascertain the achievability of user-defined value palettes. The proposed method utilizes a primal-dual approach to efficiently solve the optimization problem for a feasible choice within the palette. We verify that the dual problem exhibits concavity, enabling effective resolution through gradient ascent techniques. MAP is agnostic to the stages of model alignment, whether at the decoding or training stage, the model architecture, and the specific definitions of values used. Finally, we conducted comprehensive experimental studies to validate the practical effectiveness of MAP.

## 2 MAP: MULTI-HUMAN-VALUE ALIGNMENT PALETTE

### 2.1 PROBLEM FORMULATION

The formulation in (1) can be seen as maximizing the expected reward while imposing a regularization to minimize unnecessary deviations from the original model. This insight leads us to define a value alignment through a statistical functional constraint:

$$\mathbb{E}_{x \sim \mathcal{D}, y \sim p(\cdot | x)} r(x, y) \geq c, \tag{2}$$

which is interpreted as *the expected rewards, or realized levels, under a value preference must be at least c*. Likewise, to align $m \geq 1$ value preferences, we introduce the following **MAP problem**:

$$\min_{p \in \mathcal{P}} \mathbb{E}_{x \sim \mathcal{D}, y \sim p(\cdot | x)} D_{\mathrm{KL}}\big(p(\cdot \mid x) \,\|\, p_0(\cdot \mid x)\big) \text{ s.t. } \mathbb{E}_{x \sim \mathcal{D}, y \sim p(\cdot | x)} r_i(x, y) \geq c_i, \, \forall i = 1, \ldots, m. \tag{3}$$

We denote $\boldsymbol{c} \triangleq [c_1, \ldots, c_m]^{\mathrm{T}}$ as the *value palette*. With a solution $p$, we refer to $\mathbb{E}_{x \sim \mathcal{D}, y \sim p(\cdot | x)}(\boldsymbol{r}(x, y) \triangleq [r_1(x, y), \ldots, r_m(x, y)]^{\mathrm{T}})$ as the *realized value levels*. We write $\boldsymbol{u} \geq \boldsymbol{v}$ if the two vectors are of the same size and $u_i \geq v_i$ for each entry $i$.

**Theorem 1 (Representation of MAP solution).** The solution to the MAP problem (3) is

$$p_{\boldsymbol{\lambda}}(y \mid x) = \frac{1}{Z(x, \boldsymbol{\lambda})} p_0(y \mid x) e^{\boldsymbol{\lambda}^{\mathrm{T}} \boldsymbol{r}(x, y)}, \tag{4}$$

where $\boldsymbol{\lambda}^{\mathrm{T}} \boldsymbol{r}(x, y) = \sum_{i=1}^{m} \lambda_i r_i(x, y)$, for some $\boldsymbol{\lambda} \geq \boldsymbol{0}$. Moreover, assuming that $\boldsymbol{r}(x, y)$ is not trivially a constant on the support set of $x, y$, the above $\boldsymbol{\lambda}$ is the unique solution to the problem:

$$\max_{\boldsymbol{\lambda} \geq \boldsymbol{0}} g(\boldsymbol{\lambda}) \triangleq -\log Z(\boldsymbol{\lambda}) + \boldsymbol{\lambda}^{\mathrm{T}} \boldsymbol{c}, \tag{5}$$

where $Z(\boldsymbol{\lambda}) \triangleq \mathbb{E}_{x \sim \mathcal{D}, y \sim p_0(\cdot | x)} e^{\boldsymbol{\lambda}^{\mathrm{T}} \boldsymbol{r}(x, y)}$, and $g$ is strictly concave. As a result, we can treat $\boldsymbol{\lambda}$ in (4) as an implicit function of $\boldsymbol{c}$ and denote it as $\boldsymbol{\lambda} = \boldsymbol{\lambda}(\boldsymbol{c}) \triangleq \arg\max_{\boldsymbol{\lambda} \geq \boldsymbol{0}} g(\boldsymbol{\lambda})$.

**Remark 1 (Interpretation of $\boldsymbol{\lambda}$).** Theorem 1 establishes a one-to-one correspondence between the vectors $\boldsymbol{c}$ and $\boldsymbol{\lambda}$. The proof to the first part of Theorem 1 uses the Karush–Kuhn–Tucker (KKT) conditions, and the second part is obtained by showing that (5) is the dual problem of (3). From a decision theoretic view, the decision of $\lambda$ is based on trading off the utility term $\boldsymbol{\lambda}^{\mathrm{T}} \boldsymbol{c}$ and the "risk" term $-\log Z(\boldsymbol{\lambda})$. The latter term can be seen as a form of risk aversion, because maximizing it would penalize decisions that place a disproportionate weight on less likely, albeit highly desirable, outcomes. Practically, the expectation $\mathbb{E}_{x \sim \mathcal{D}, y \sim p_0(\cdot | x)}$ can be easily approximated using a sample average from a dataset generated under $p_0$, allowing the dual problem (5) to be numerically solved. This computational aspect is further explored in Section 2.3.

**Remark 2 (Choice of Value Palette $\boldsymbol{c}$).** We outline interpretable methods for selecting the value palette $\boldsymbol{c}$. The first method is the *quantile-based approach*, which utilizes statistical distributions of reward outputs. For any reward function, input prompts $x \in \mathcal{D}$, and model-generated outputs $y \sim p_0(\cdot \mid x)$, the mapping $(x, y) \mapsto \boldsymbol{r}(x, y)$ defines a random variable, denoted by $\mathcal{R}(\mathcal{D}, p_0)$. The expectation of $\mathcal{R}(\mathcal{D}, p_0)$ is the realized value under the model $p_0$. Users can set each component of $\boldsymbol{c}$ to correspond to a quantile of $\mathcal{R}(\mathcal{D}, p_0)$. For example, for the first value, setting $c_1$ to the 80% quantile positions the realized level of the aligned model $p$ within the upper 20% of expected outcomes. For practical implementation, we propose using a small pilot dataset generated from the

original model as Monte Carlo samples to empirically estimate the quantiles of $\mathcal{R}(\mathcal{D}, p_0)$. That is, we sample $x_i \sim \mathcal{D}, y_i \sim p_0(\cdot \mid x_i)$ to create $\{r(x_i, y_i) : i = 1, \ldots, n\}$ ($n = 2000$ in our experiments). This pilot data can serve various palette choices. We provide further remarks on two other approaches, one based on classifiers and another based on automatic adjustment in Appendix B.4.

**Remark 3 (Robustness of MAP under Mis-specified Rewards).** We conducted a theoretical analysis of robustness against mis-specification in Appendix B.1. Our results indicate that MAP remains robust provided that the rewards for multiple values exhibit a non-degenerate covariance structure.

## 2.2 REALIZABLE VALUE LEVELS OF THE MAP PROBLEM AND PARETO FRONTIER

We first show that the MAP problem can be written as the original alignment problem (1) with a particular reward function that is simply a linear combination of individual rewards.

**Theorem 2 (Solution of MAP).** The solution to the MAP problem (3) is the same of the problem (1) with $R(x, y) \triangleq \beta \cdot \lambda(c)^{\mathsf{T}} r(x, y)$, where $\lambda(c)$ was introduced in Theorem 1.

This prompts the natural question of whether MAP limits the breadth of realizable value levels compared to those achievable under (1) with arbitrarily chosen $R$. Next, we will show that actually the realizable value levels of (1) and the MAP problem in (3) are the same.

Given the reference distribution $p_0$ and any specific reward functions $R$, the solution $p$, if feasible, depends solely on $R$. To illustrate this dependency, we denote $p$ as $p_R$. Let $\mathfrak{F}_{\mathrm{RLHF}}(p_0)$ represent the range of $R$ that admits feasible solutions to the RLHF problem, essentially those with valid probability densities. The realizable value levels under the RLHF problem are defined as:

$$\mathcal{V}_{\mathrm{RLHF}}(p_0) \triangleq \left\{ \mathbb{E}_{x \sim \mathcal{D}, y \sim p_R(\cdot|x)} r(x, y) : R \in \mathfrak{F}_{\mathrm{RLHF}}(p_0) \right\}.$$

For multiple reward functions $r_1, \ldots, r_m$, we consider a specific class of $R$ comprising various non-negative linear combinations, and define their realizable value levels similarly:

$$\mathcal{V}_{\mathrm{RLHF}}(r_1, \ldots, r_m; p_0) \triangleq \left\{ \mathbb{E}_{x \sim \mathcal{D}, y \sim p_R(\cdot|x)} r(x, y) : R \triangleq \sum_{i=1}^{m} \rho_i r_i \in \mathfrak{F}_{\mathrm{RLHF}}(p_0), \ \rho_i \geq 0 \right\}.$$

In the MAP problem, given $p_0$ and $r$, the solution $p$, if feasible, depends only on the user-specified value palette $c$. To emphasize this relationship, we denote $p$ as $p_c$. Let $\mathcal{C}_{\mathrm{MAP}}(r_1, \ldots, r_m; p_0)$ denote the range of $c$ that admits feasible solutions to the MAP problem. We further consider the realized value levels of all feasible solutions under various $c$, defined as:

$$\mathcal{V}_{\mathrm{MAP}}(r_1, \ldots, r_m; p_0) \triangleq \left\{ \mathbb{E}_{x \sim \mathcal{D}, y \sim p_c(\cdot|x)} r(x, y) : c \in \mathcal{C}_{\mathrm{MAP}}(r_1, \ldots, r_m; p_0) \right\}.$$

**Theorem 3 (Equivalent realizable value levels).** For any original generative model $p_0$, we have

$$\mathcal{V}_{\mathrm{MAP}}(r_1, \ldots, r_m; p_0) = \mathcal{V}_{\mathrm{RLHF}}(p_0) = \mathcal{V}_{\mathrm{RLHF}}(r_1, \ldots, r_m; p_0). \tag{6}$$

Theorem 3 establishes that the realizable value levels by MAP equate to those in the original alignment problem (1) using a specific reward function – a linear combination of individual rewards. This proves that linear combinations of individual reward functions can sufficiently capture the entire Pareto Frontier. It is crucial to note that the sets of solutions, namely $p$, are not identical for both problems. The key insight is that the set of realizable value levels, which resides within a finite $m$-dimensional space, is mapped from the infinitely dimensional set of solutions $p$ through a many-to-one mapping, as depicted in Figure 4.

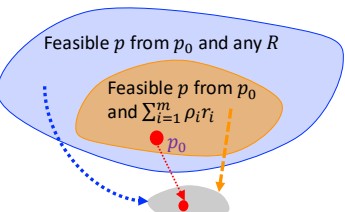

Figure 4: Illustration of Theorem 3.

We denote all feasible value palettes $\mathcal{C}_{\mathrm{MAP}}(r_1, \ldots, r_m; p_0)$ simply as $\mathcal{C}$. The following theorem explores the characteristics of this set.

**Theorem 4.** The following properties of $\mathcal{C}$ hold.

(i) If $c \in \mathcal{C}$, for any $c'$ such that $c' \leq c$, we have $c' \in \mathcal{C}$.

(ii) For any $c'$ such that $c' \leq c_0 \triangleq \mathbb{E}_{x \sim \mathcal{D}, y \sim p_0(\cdot|x)} r(x, y)$, the value level of the original model, we have the corresponding $\lambda(c') = 0$.

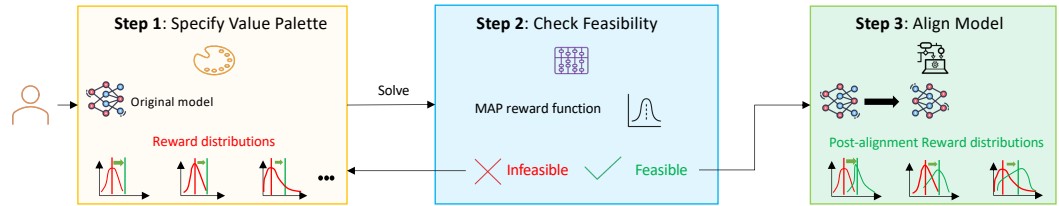

Figure 5: Overview of the MAP procedure, which 1) lets a user specify desirable levels of expected levels for all values of interest, referred to as a *Value Palette*, 2) checks whether the specified palette admits a feasible solution to model alignment, 3) actually aligns the model by using a single MAP-guided reward function.

---

**Algorithm 1** MAP Procedure

---

**Input:** Multi-dimensional reward functions $\boldsymbol{r}$, original generative model $p_0(\cdot \mid x)$ with $x \sim \mathcal{D}$.

——————————————————**Step 1**: Specify value palette ——————————————————
- Specify $\boldsymbol{c}$ (e.g., based on Remark 2).

——————————————————**Step 2**: Check feasibility ——————————————————
- Attempt to solve Problem (7) with updates in (8).
  **If infeasible**: Suggest alternative $\boldsymbol{c}$ (Remark 2).

  **Else**: Obtain $\boldsymbol{\lambda}$ and define reward function $R \triangleq \boldsymbol{\lambda}^{\mathrm{T}}\boldsymbol{r}$.

——————————————————**Step 3**: Align model ——————————————————
- Decoding (D) option: For a prompt $x$, generate $y^{(1)}, \ldots, y^{(m)}$, and sample $y = y^{(i)}$ using the Softmax of $R(x, y^{(i)})$ as Multinomial probability.
- Finetuning (F) option: Apply Proximal Policy Optimization (PPO) with $R$ to fine-tune $p_0$ into $\hat{p}$. For a prompt $x$, generate $y \sim \hat{p}(\cdot \mid x)$.

**Output:** The generated $y$.

---

(iii) Let $B_i \triangleq \operatorname{ess\,sup}_{x \sim \mathcal{D}, y \sim p_0(\cdot \mid x)} r_i(x, y)$, where the supreme is under the original model $p_0$. Then, $\mathcal{C} \subseteq (-\infty, B_1] \times \cdots \times (-\infty, B_m]$. That is, there is no feasible solution if $c_i > B_i$ for some $i$.

(iv) Let $p_{\boldsymbol{\lambda}}(y \mid x) \triangleq p_0(y \mid x)e^{\boldsymbol{\lambda}^{\mathrm{T}}\boldsymbol{r}(x,y)}/Z(\boldsymbol{\lambda}, x)$ be the $\boldsymbol{\lambda}$-adjusted distribution of $p_0$, where $Z(\boldsymbol{\lambda}, x) \triangleq \int_{y'} p_0(y' \mid x)e^{\boldsymbol{\lambda}^{\mathrm{T}}\boldsymbol{r}(x,y')}dy'$ is the normalizing constant. Treating the optimal solution $\boldsymbol{\lambda} = \boldsymbol{\lambda}(\boldsymbol{c})$ as a function of $\boldsymbol{c}$, we have $\frac{d\boldsymbol{\lambda}(\boldsymbol{c})}{d\boldsymbol{c}} = -V^{-1}$, where $V \triangleq \mathit{Var}_{\boldsymbol{\lambda}}(\boldsymbol{r}(x, y))$ is the covariance matrix of $\boldsymbol{r}(x, y)$ under the distribution $x \sim \mathcal{D}, y \mid x \sim p_{\boldsymbol{\lambda}}(\cdot \mid x)$.

The result in (iv) can be used to check if a small increase in $\boldsymbol{c}$, e.g., changing $c_1$ to $c_1 + \delta$ for a small $\delta > 0$, will cause $\boldsymbol{\lambda}$ to change in a way that remains feasible.

## 2.3 COMPUTATIONAL SOLUTION TO MAP THROUGH A PRIMAL-DUAL APPROACH

In this section, we introduce a practical framework to solve MAP problem defined in (3), illustrated in Figure 5 and Algorithm 1. Once a user defines a value palette according to Remark 2, in Step 2, we derive $\boldsymbol{\lambda}$ from $\boldsymbol{c}$ as per Theorem 1. We then approximate Problem (5) as follows:

$$\max_{\boldsymbol{\lambda} \geq \boldsymbol{0}} g_n(\boldsymbol{\lambda}) \triangleq -\log \frac{1}{n} \sum_{j=1}^{n} e^{\boldsymbol{\lambda}^{\mathrm{T}}\boldsymbol{r}(x_j, y_j)} + \boldsymbol{\lambda}^{\mathrm{T}}\boldsymbol{c}, \tag{7}$$

where the dataset $\{(x_j, y_j)_{j=1}^n\}$ serves as a finite-sample approximation of the distribution $p_0(x, y)$ with $y_1, \ldots, y_n$, generated conditional on respective prompts $x_1, \ldots, x_n$. Lemma 1 in the Appendix confirms that both the original problem (5) and its approximation (7) are concave, allowing for the use of gradient ascent to efficiently find the optimal $\boldsymbol{\lambda}$:

$$\boldsymbol{\lambda} \leftarrow \boldsymbol{\lambda} + \alpha \frac{d}{d\boldsymbol{\lambda}}g(\boldsymbol{\lambda}) = \boldsymbol{\lambda} + \alpha\big(-\mathrm{Softmax}(\boldsymbol{\lambda}^{\mathrm{T}}\boldsymbol{r}_{1:n}) \cdot \boldsymbol{r}_{1:n}^{\mathrm{T}} + \boldsymbol{c}\big) \tag{8}$$

where $\boldsymbol{r}_{1:n} \triangleq [\boldsymbol{r}(x_1, y_1), \ldots, \boldsymbol{r}(x_n, y_n)]$. In practical implementations, we applied a change of variable $\boldsymbol{\tau} \triangleq \log \boldsymbol{\lambda}$ to ensure the constraints $\boldsymbol{\lambda} \geq \boldsymbol{0}$ are satisfied. Not all users' palette choices are feasible. If the updates in (8) do not converge, this indicates that the chosen $\boldsymbol{c}$ is infeasible. In such

cases, MAP can adjust the value palette $c$ automatically, which we elaborate in Appendix B.2. Also, because the above optimization problem is concave and the data can be pre-computed using a pilot set of Monte Carlo samples from the original model, solving the problem is efficient and does not depend on the complexity of the models to align. More discussions are included in Appendix B.3.

In Step 3, we implement $p(y \mid x) \propto p_0(y \mid x)e^{\boldsymbol{\lambda}^{\mathrm{T}}\boldsymbol{r}(x,y)}$ through two practical approaches during experiments. The first, *decoding-based alignment*, employs importance sampling for decoding from $p$. Specifically, for any input $x$, we generate $k$ samples from $p_0$, labeled $y_1, \ldots, y_k$, and select one via Multinomial sampling with weights proportional to $\exp(\boldsymbol{\lambda}^{\mathrm{T}}\boldsymbol{r}(x, y_i))$ for each sample $i$. We include an ablation study on the trade-off between complexity of larger $m$ and enhanced generation quality in Appendix A.4. The second approach, *finetuning-based alignment*, fine-tunes a model $p_w$ with neural weights $w$ based on the reward function $(x, y) \mapsto \boldsymbol{\lambda}^{\mathrm{T}}\boldsymbol{r}(x, y)$. Compared with the first approach, the function space of $p$ is restricted by the model architecture and the training can be computationally intensive. However, it can reduce the inference-stage complexity by only decoding one sentence for each prompt. An experimental study on this comparison is included in Appendix A.5.

## 2.4 SIMULTANEOUS VERSUS SEQUENTIAL OR GREEDY ALIGNMENT

To align a model with a specified value palette $c$, a natural baseline method is sequentially aligning individual values, namely we sequentially update the model to align each value to an entry in $c$ until all values are addressed. This section compares this method with the MAP procedure.

**Sequential alignment algorithm**. Suppose we have aligned with one value palette $c_{(\ell-1)}$ and then update with $c_{(\ell)}$, $\ell = 1, 2, \ldots$. For notational convenience, let the initial $c_{(0)}$ be the realized level of the original model $p_0$. At the end of the $\ell$-th alignment, the aligned distribution can be expressed as $p_{(\ell)}(y \mid x) \propto p_0(y \mid x) \cdot \exp(\boldsymbol{\lambda}_{(\ell)}^{\mathrm{T}}\boldsymbol{r}(x, y))$ for some vector $\boldsymbol{\lambda}_{(\ell)}$.

Given the value palettes $\{c_{(\ell)}\}_{\ell=1,2,\ldots}$, we recursively obtain $\boldsymbol{\lambda}_{(\ell)}$ from $\boldsymbol{\lambda}_{(\ell-1)}$. Like the problem (5), the alignment objective at the beginning of the $\ell$-th alignment is:

$$\max_{\boldsymbol{\lambda} \geq \mathbf{0}} g_{(\ell)}(\boldsymbol{\lambda}) \triangleq -\log \mathbb{E}_{x \sim \mathcal{D}, y \sim p_{(\ell-1)}(\cdot \mid x)} e^{\boldsymbol{\lambda}^{\mathrm{T}}\boldsymbol{r}(x,y)} + \boldsymbol{\lambda}^{\mathrm{T}} c_{(\ell)}. \tag{9}$$

Like the problem (7), we can numerically solve (9) by addressing the problem:

$$\max_{\boldsymbol{\lambda} \geq \mathbf{0}} g_{(\ell)}(\boldsymbol{\lambda}) \triangleq -\log\big(\mathrm{Softmax}(\boldsymbol{\lambda}_{(\ell-1)}^{\mathrm{T}}\boldsymbol{r}_{1:n}) \cdot e^{\boldsymbol{r}_{1:n}^{\mathrm{T}}\boldsymbol{\lambda}}\big) + \boldsymbol{\lambda}^{\mathrm{T}} c_{(\ell)}. \tag{10}$$

**Connection between Sequential and Simultaneous Alignment**. The problem (9) establishes a connection between the $(\ell-1)$-th and $\ell$-th alignment. Recall that $m$ denotes the number of values or the dimension of $c$. In a canonical setting where one value is aligned at a time, define $\{c_{(\ell)}\}_{\ell=1,2,\ldots}$ by: $c_{(\ell)} = [-\infty, \ldots, c_\ell, -\infty, \ldots]$, for $\ell \leq m$, where the $\ell$-th element is the same as that of $c$, and the others are trivially negative infinity. For $\ell > m$, we can re-align the first value, and so on. Namely, we replace the above $c_\ell$ in $c_{(\ell)}$ with $c_{\ell \bmod m}$. The following result shows this sequential, iterative alignment process converges to the joint alignment using MAP.

**Theorem 5.** Let $p_{(0)} \triangleq p_0$, $p_{(1)}$, $p_{(2)}$, $\ldots$ be the sequence of distributions obtained by sequentially aligning the original model according to the single-value MAP objective:

$$\min_{p \in \mathcal{P}} \mathbb{E}_{x \sim \mathcal{D}, y \sim p(\cdot \mid x)} D_{\mathrm{KL}}\big(p(\cdot \mid x) \,\|\, p_{(\ell-1)}(\cdot \mid x)\big) \ \text{s.t.} \ \mathbb{E}_{x \sim \mathcal{D}, y \sim p(\cdot \mid x)} r_{\ell'}(x, y) \geq c_{\ell'},$$

where $\ell' = (\ell \bmod m)$. Assuming $\boldsymbol{r}(x, y)$ is not trivially a constant on the support set of $(x, y)$, this sequence weakly converges to $p_{\boldsymbol{\lambda}(\boldsymbol{c})}$, which is the solution to the MAP problem in (4).

The proof of Theorem 5 demonstrates that the sequential alignment process fundamentally operates as a coordinate-wise gradient ascent algorithm. When we align subsets of values in each iteration, the process can equivalently function as a block-wise ascent. Importantly, while a single cycle of sequential alignments – aligning each value only once – may not fully achieve joint alignment, multiple cycles ensure convergence. This aspect is particularly significant in scenarios where constraints such as limited GPU memory restrict the simultaneous loading of numerous reward models. Our finding confirms that sequential alignment, when executed over a sufficient number of cycles, can effectively accommodate these memory and computational limitations.

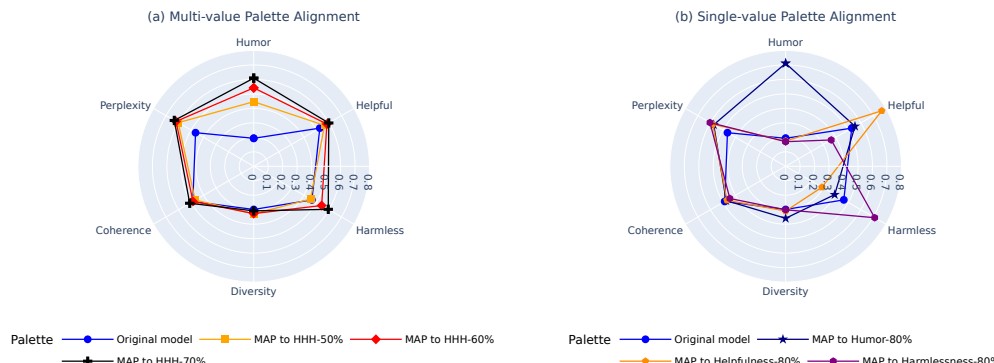

Figure 6: Radar plots showing the alignment of OPT-1.3B with (a) multi-value palettes given by 50%, 60%, and 70% quantiles of the original model's reward distributions, and (b) single-value palettes at the 80% quantile.

## 3 EXPERIMENTAL STUDY

### 3.1 EXPERIMENTAL SETUP

We generate prompts from two data sources: Anthropic harmless data (Bai et al., 2022), which includes human requests delineated between the tags "Human:" and "Assistant:", and IMDB data (Maas et al., 2011) from which we retain movie reviews exceeding 30 characters in length. For backbone models, we employ OPT-1.3B (Zhang et al., 2022) and Llama2-7B-chat (Touvron et al., 2023), which have demonstrated robust language modeling capabilities in previous assessments. We focus on several values for alignment: Humor, Positiveness, Harmlessness, Helpfulness, Diversity, Coherence, and Perplexity. The Humor reward is assessed using the logits from a humor detection model (Dhiab, 2023), while Positiveness uses a DistilBERT model trained on IMDB sentiment analysis (Lvwerra, 2021). Harmlessness and Helpfulness are evaluated through two GPT-2 models equipped with a value head fine-tuned to predict these attributes (Yang et al., 2024). Diversity is measured by the lexical variety within sentences, calculated through the proportion of unique $n$-grams ($n = 2, 3, 4$) and their composite score (Zhang et al., 2020). Coherence is determined by the semantic similarity of sentences within a context, using a supervised SimCSE BERT-based model that captures sentence embeddings to assess textual coherence (Gao et al., 2021).

### 3.2 EFFECTIVENESS OF MAP FOR SIMULTANEOUS MULTI-VALUE ALIGNMENT

For the conversational task, we use Anthropic data as input prompts $x$ to align the OPT-1.3B model across six dimensions: Humor, Harmlessness, Helpfulness, Diversity, Coherence, and Perplexity. Here, the perplexity is calculated as negative log of the standard perplexity so that the larger the better. We tested seven different value palettes for alignment, considering both decoding-based and finetuning-based implementations. For example, the HHH-80% palette aligns the first three values to the 80th percentile of their respective distributions, while maintaining the last three values. This approach aims to enhance the model's output to closely resemble human-like interaction standards without compromising the intrinsic qualities of the generated content. The results are summarized in Table 1. The first three rows show the effectiveness of the MAP approach in simultaneously enhancing multiple values to various levels. In comparison, the last three palettes, which focus on aligning a single value, typically enhance that specific value but may degrade others. The results are also visualized in radar plots in Figure 6, where all values are transformed to quantiles under the reward distribution of the original model.

Additional experiments, including an ablation study with the Llama2-7B-chat model, are detailed in Appendix A.1. Notably, the HHH-80% palette was determined to be feasible by Algorithm 1 Step 2 and its results are therefore included. This indicates that the Llama2-7B-chat model, which has a larger complexity than OPT-1.3B, allows for more extensive multi-value alignment. We also explored a sentiment-controlled open generation task using a random trunk of IMDB data as input prompts to align the OPT-1.3B model. The results are summarized in Table 2.

### 3.3 COMPARISON WITH OTHER BASELINE APPROACHES

To demonstrate the effectiveness of MAP in reliably aligning multiple values, we conduct an experiment showing that MAP can identify desirable outcomes with randomly sampled $\boldsymbol{\lambda}$. According to

Table 1: Results of aligning the OPT-1.3B model with Anthropic conversational data using diverse value palettes, evaluated in terms of expected rewards (or realized value levels). Scenarios include: 1) enhancing the first three values – Humor, Harmlessness, Helpfulness – to the $x$th percentile of their respective reward distributions under the original model ("HHH-$x$%") while maintaining the last three values – Diversity, Coherence, and Perplexity – using MAP, and 2) aligning individual values independently ("Humor-80%", "Helpfulness-80%", "Harmlessness-80%"). Results include the optimized weight vector $\boldsymbol{\lambda}$, targeted value palettes, and realized value levels. Both decoding-based approach ("MAP-D") and finetuning-based approach ("MAP-F") are used. The palette "HHH-80%" was found infeasible, as confirmed in Step 2 of Algorithm 1.

| | Humor↑ | Helpfulness↑ | Harmlessness↑ | Diversity↑ | Coherence↑ | Perplexity↑ |
|---|---|---|---|---|---|---|
| **Original model** | 2.07 | -1.47 | 0.25 | 0.88 | 0.43 | -3.34 |
| **HHH-50%** ($\boldsymbol{\lambda} = [2.53, 0.23, 0.28, 0.02, 0.05, 0.05]$) | | | | | | |
| MAP-D | **2.44** | **-1.38** | 0.21 | **0.88** | **0.43** | **-3.17** |
| MAP-F | **2.20** | -1.80 | **0.66** | 0.86 | 0.41 | **-2.80** |
| **HHH-60%** ($\boldsymbol{\lambda} = [6.30, 0.83, 0.93, 0.01, 0.03, 0.03]$) | | | | | | |
| MAP-D | **2.48** | **-1.33** | **0.48** | **0.88** | **0.43** | **-3.15** |
| MAP-F | **2.47** | -2.26 | **0.50** | **0.89** | 0.26 | -3.49 |
| **HHH-70%** ($\boldsymbol{\lambda} = [12.77, 1.53, 1.69, 0.01, 0.02, 0.02]$) | | | | | | |
| MAP-D | **2.49** | **-1.29** | **0.66** | **0.88** | **0.45** | **-3.14** |
| MAP-F | **2.17** | -2.28 | **0.97** | 0.82 | 0.10 | -5.04 |
| **Humor-80%** ($\boldsymbol{\lambda} = [16.44, -, -, -, -, -]$) | | | | | | |
| MAP-D | **2.52** | **-1.42** | 0.01 | **0.89** | **0.43** | **-3.21** |
| MAP-F | **2.08** | -2.49 | **0.39** | 0.79 | 0.09 | -5.85 |
| **Helpfulness-80%** ($\boldsymbol{\lambda} = [-, 0.72, -, -, -, -]$) | | | | | | |
| MAP-D | 1.99 | **-0.75** | -0.35 | **0.88** | **0.43** | **-3.20** |
| MAP-F | 2.02 | **-0.66** | -0.58 | **0.88** | 0.41 | **-2.73** |
| **Harmlessness-80%** ($\boldsymbol{\lambda} = [-, -, 1.27, -, -, -]$) | | | | | | |
| MAP-D | 1.97 | -1.86 | **0.97** | **0.88** | 0.42 | **-3.17** |
| MAP-F | 2.05 | -2.02 | **0.94** | 0.87 | 0.40 | **-2.63** |

Table 2: Results of aligning the OPT-1.3B model for the open-generation task with IMDB data using diverse value palettes, evaluated in terms of expected rewards (or realized value levels). Scenarios include: 1) enhancing the first three values – Positiveness, Harmlessness, Helpfulness – to the $x$th percentile of their respective reward distributions under the original model ("PHH-$x$%") while maintaining the last three values – Diversity, Coherence, and Perplexity – using MAP, and 2) aligning individual values independently ("Positiveness-80%", "Helpfulness-80%", and "Harmlessness-80%").

| | Positiveness↑ | Helpfulness↑ | Harmlessness↑ | Diversity↑ | Coherence↑ | Perplexity↑ |
|---|---|---|---|---|---|---|
| **Original model** | 0.52 | -1.53 | 0.58 | 0.88 | 0.24 | -3.36 |
| **PHH-50%** ($\boldsymbol{\lambda} = [0.24, 0.08, 0.12, 0.03, 0.07, 0.07]$) | | | | | | |
| MAP-D | **0.62** | **-1.40** | 0.57 | **0.88** | **0.24** | **-3.19** |
| MAP-F | **0.55** | **-1.50** | 0.49 | **0.88** | 0.23 | **-2.80** |
| **PHH-60%** ($\boldsymbol{\lambda} = [2.23, 0.41, 0.79, 0.01, 0.03, 0.04]$) | | | | | | |
| MAP-D | **0.91** | **-1.16** | **0.67** | **0.89** | **0.25** | **-3.13** |
| MAP-F | **0.88** | **-0.69** | 0.38 | 0.87 | **0.24** | **-2.46** |
| **PHH-70%** ($\boldsymbol{\lambda} = [3.83, 0.90, 1.48, 0.01, 0.02, 0.03]$) | | | | | | |
| MAP-D | **0.93** | **-1.10** | **0.74** | **0.89** | **0.24** | **-3.13** |
| MAP-F | **0.94** | **-0.33** | 0.24 | 0.80 | 0.18 | **-2.64** |
| **PHH-80%** ($\boldsymbol{\lambda} = [9.77, 1.42, 2.27, 0.00, 0.01, 0.03]$) | | | | | | |
| MAP-D | **0.93** | **-1.06** | **0.64** | **0.89** | **0.25** | **-3.14** |
| MAP-F | **0.95** | **-1.22** | 0.24 | 0.73 | 0.18 | **-2.99** |
| **Positiveness-80%** ($\boldsymbol{\lambda} = [10.98, -, -, -, -, -]$) | | | | | | |
| MAP-D | **0.94** | **-1.10** | 0.47 | **0.89** | **0.25** | **-3.17** |
| MAP-F | **0.92** | **-1.27** | 0.37 | 0.73 | 0.17 | **-3.00** |
| **Helpfulness-80%** ($\boldsymbol{\lambda} = [-, 0.95, -, -, -, -]$) | | | | | | |
| MAP-D | **0.56** | **-0.86** | 0.29 | **0.89** | 0.23 | **-3.17** |
| MAP-F | **0.55** | **-1.28** | 0.34 | **0.88** | 0.23 | **-2.83** |
| **Harmlessness-80%** ($\boldsymbol{\lambda} = [-, -, 1.43, -, -, -]$) | | | | | | |
| MAP-D | **0.53** | -1.80 | **1.21** | **0.89** | **0.24** | **-3.17** |
| MAP-F | 0.47 | -1.77 | **0.91** | **0.89** | 0.23 | **-2.62** |

Theorem 4, we sample $\boldsymbol{\lambda}$ randomly from the range $(\boldsymbol{c_0}, \boldsymbol{B})$ and retain the feasible ones according to MAP's feasibility check. From these, we select $\boldsymbol{\lambda}$ vectors with a bounded $\ell_1$-norm less than 6. We compare this with a standard MORL approach where $\boldsymbol{\lambda}$ is randomly generated from $s \cdot \boldsymbol{u}$, where $s$ is

uniformly sampled from $(0, 6)$ and $\boldsymbol{u}$ is uniformly sampled from the probability simplex. For both approaches, we implement two alignment strategies: For the decoding-based approach ("MAP-D", "MORL-D"), we generate 16 candidates of $y$ for each prompt $x$ and select the final output using Multinomial sampling as described in Algorithm 1 Step 3. For the finetuning-based approach ("MAP-F", "MORL-F"), we apply PPO and the calculated MAP reward function $R$ to fine-tune the original model. We evaluate the outcomes using the "test" split of our task data, assessing both the expected reward and win rate against the original model based on the reward functions.

Additionally, we compare MAP with the DPO approach that uses various mixtures of preference data, and DPO-Soup, which applies linear interpolation of DPO-trained, value-specific models. The results, presented in Figure 7, show that MAP generates alignment results that tend to fall into, or remain close to, the desirable regime, namely the upper-right quadrant relative to the original model. In contrast, the other approaches experience more severe trade-offs and rarely fall into the desirable regime. Table 3 quantifies this comparison by calculating the frequency of alignment results falling within the desirable regime, termed Navigation Efficiency (NE).

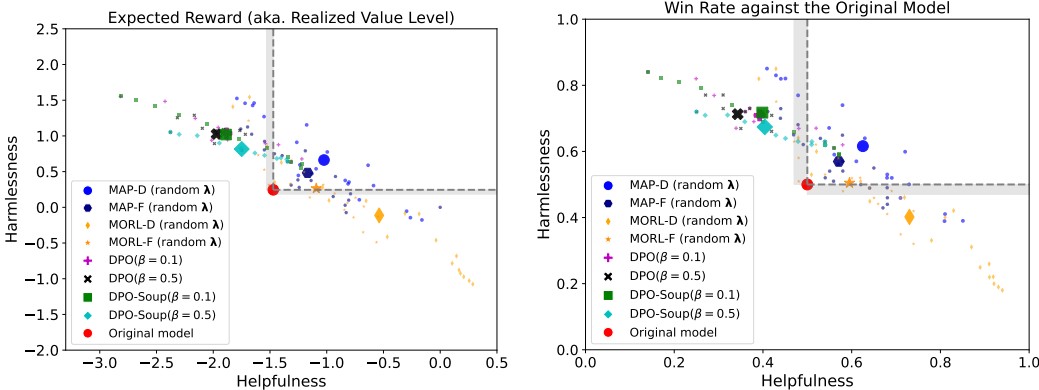

Figure 7: Comparison of various methods for aligning the OPT-1.3B model towards Helpfulness and Harmlessness. The performance is evaluated by Expected Reward (left plot) and Win Rate (right plot) against the original model. In MAP-D, we used MAP to obtain 20 $\boldsymbol{\lambda}$'s based on randomly sampled palettes, and test data is generated using decoding-stage resampling. MAP-F is similar to MAP-D, except it uses PPO to fine-tune the model. MORL-D and MORL-F follow the same structure as MAP but with randomly sampled $\boldsymbol{\lambda}$. DPO($\beta$) applies DPO with a regularization parameter $\beta$ to a mixture of Anthropic Helpful and Harmless datasets, with mixing ratios ranging from 0 to 1. DPO-Soup($\beta$) is a weighted average of the model parameters from DPO($\beta$) obtained at mixing ratios from 0 to 1. Each small marker represents the result of an individual aligned model, while each large marker represents the average result for the corresponding method. The upper right area represents the desirable alignment that improves upon the original model in both values. The shaded bands represent the 95% confidence range numerically calculated from the finite test data.

Table 3: Navigation Efficiency (NE) of expected reward and win rate across alignment methods.

|  | **MAP-D** | **MAP-F** | MORL-D | MORL-F | DPO(0.1) | DPO(0.5) | DPO-Soup(0.1) | DPO-Soup(0.5) |
|---|---|---|---|---|---|---|---|---|
| NE of expected reward | **55%** | **52%** | 21% | 35% | 27% | 9% | 45% | 36% |
| NE of win rate | **60%** | **70%** | 21% | 55% | 36% | 9% | 45% | 45% |

## 4 CONCLUSION

The proposed MAP offers a structured approach to aligning multiple human values, enabling precise adjustments to meet diverse user preferences. Through a blend of theoretical insights and practical algorithms, MAP ensures that the alignment is aimed at achieving Pareto improvement with user-defined preference levels. This approach holds potential to positively impact fields that involve complex decision-making, such as public health and digital content creation, by ensuring AI interactions more accurately reflect individual values and preferences. Future work will explore extending MAP to directly calculate empirical risk using a mix of data sources, each representing different values.

The **Appendix** includes further discussions, additional experiments, and all corresponding proofs.

ACKNOWLEDGEMENT

The work was supported in part by the 3M Science and Technology Graduate Fellowship, the Samsung Global Research Outreach Award, the Army Research Office Early Career Program under grant number W911NF2310315, and National Science Foundation CAREER Program under grant number 2338506.

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

# A ADDITIONAL EXPERIMENTAL RESULTS

## A.1 ABLATION STUDY WITH LARGER MODEL

Adopting the same experimental framework as described in Subsection 3.2, we substituted the OPT-1.3B model with the Llama2-7B-chat model. The results are summarized in Table 4. Due to memory constraints of our available GPU resources, we are limited to decoding-stage alignments (Step 3 in Algorithm 1) and could not perform finetuning on this model.

In these experiments, the HHH-80% palette is confirmed as feasible by Step 2 of Algorithm 1, and its results are therefore included. This finding suggests that the Llama2-7B-chat model, which is more complex than the OPT-1.3B, provides greater flexibility for multi-value alignment.

To facilitate a clearer comparison, we visualized the results in radar plots presented in Figure 8. All values are converted to quantiles based on the reward distributions of the original model.

Table 4: Results of aligning the Llama2-7B-chat model with the Anthropic prompt data using diverse value palettes, evaluated in terms of expected rewards (or realized value levels). Scenarios include: 1) enhancing the first three values – Humor, Harmlessness, Helpfulness – to the $x$th percentile of their respective reward distributions under the original model ("HHH-$x$%") while maintaining the last three values – Diversity, Coherence, and Perplexity – using MAP, and 2) aligning individual values independently ("Humor-80%", "Helpfulness-80%", "Harmlessness-80%"). Results include the optimized weight vector $\boldsymbol{\lambda}$, targeted value palettes, and realized value levels. Notably, **in contrast with Table 1, the "HHH-80%" palette was determined to be feasible** by Algorithm 1 Step 2 and is therefore included.

| | Humor↑ | Helpfulness↑ | Harmlessness↑ | Diversity↑ | Coherence↑ | Perplexity↑ |
|---|---|---|---|---|---|---|
| **Original model** | 0.60 | -1.01 | 1.25 | 0.85 | 0.52 | -1.38 |
| **HHH-50%** ($\boldsymbol{\lambda}$ = [0.27, 0.20, 0.21, 0.10, 0.04, 0.09]) | | | | | | |
| MAP-D | **0.98** | -1.06 | **1.36** | 0.84 | **0.53** | **-1.37** |
| **HHH-60%** ($\boldsymbol{\lambda}$ = [0.85, 0.78, 0.79, 0.01, 0.02, 0.07]) | | | | | | |
| MAP-D | **1.56** | **-0.93** | **1.47** | 0.84 | **0.53** | **-1.37** |
| **HHH-70%** ($\boldsymbol{\lambda}$ = [2.02, 1.39, 1.50, 0.01, 0.02, 0.09]) | | | | | | |
| MAP-D | **2.01** | **-0.87** | **1.57** | 0.83 | **0.55** | **-1.37** |
| **HHH-80%** ($\boldsymbol{\lambda}$ = [5.94, 2.43, 2.92, 0.01, 0.01, 0.15]) | | | | | | |
| MAP-D | **2.17** | **-0.93** | **1.68** | 0.82 | **0.55** | **-1.37** |
| **Humor-80%** ($\boldsymbol{\lambda}$ = [2.89, −, −, −, −, −]) | | | | | | |
| MAP-D | **2.21** | -1.26 | 1.06 | 0.82 | **0.55** | -1.47 |
| **Helpfulness-80%** ($\boldsymbol{\lambda}$ = [−, 0.69, −, −, −, −]) | | | | | | |
| MAP-D | 0.50 | **-0.39** | 0.94 | **0.86** | 0.53 | **-1.36** |
| **Harmlessness-80%** ($\boldsymbol{\lambda}$ = [−, −, 0.99, −, −, −]) | | | | | | |
| MAP-D | 0.47 | -1.33 | **1.91** | **0.86** | 0.50 | **-1.34** |

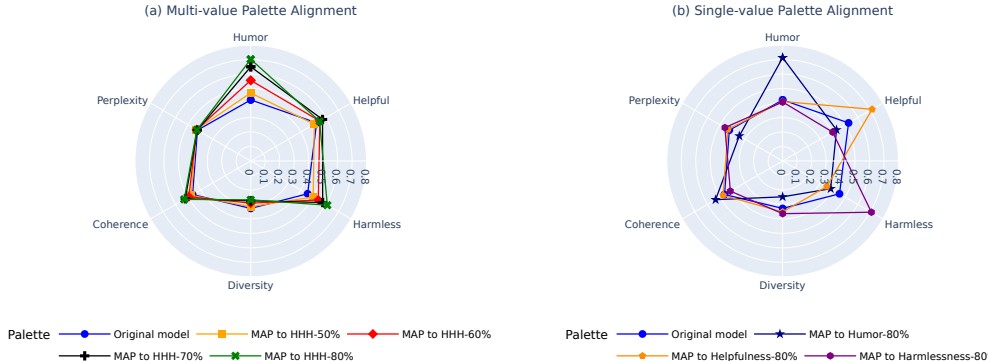

Figure 8: Radar plots showing the alignment of Llama2-7B-chat with (a) multi-value palettes as determined by 50%, 60%, and 70% quantiles of the original models' reward distributions (numerically computed using generated data), and (b) single-value palettes at the 80% quantile.

## A.2 Simultaneous Versus Sequential Alignment

To corroborate Theorem 5 in Section 2.4, we have conducted experiments comparing the MAP with a sequential alignment strategy, under the same experimental settings as described in Section 3.2. Specifically, we align the OPT-1.3B model for a conversational task using Anthropic data. We implement sequential alignment through one and five rounds. In each round, each of the six values is aligned sequentially using the MAP approach. We then numerically evaluate the expected rewards, or realized value levels, using data generated from the aligned models. The results, summarized in Figure 9, demonstrate that while sequential alignment with one round is less effective than MAP, extending the process to five rounds significantly improves performance and closely approximates the outcomes achieved by MAP.

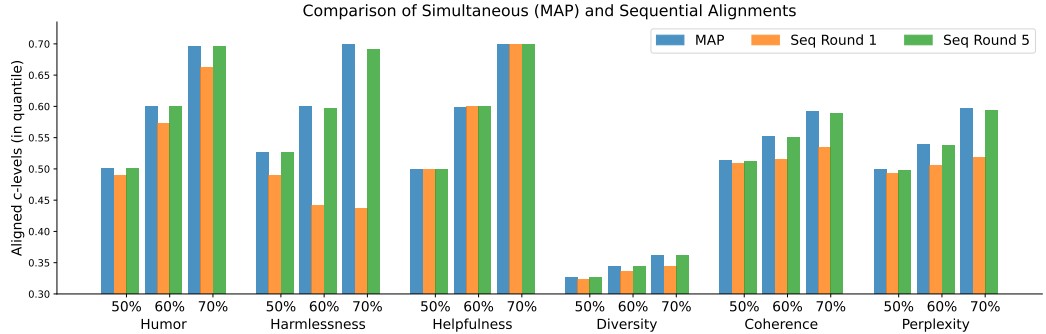

Figure 9: Comparison of value levels achieved by MAP and sequential alignments across different rounds, using the same experimental settings as described in Section 3.2.

## A.3 Hyperparameters in Data Generation and Model Training

Our experiments were conducted using a single Nvidia A100 GPU. For data generation, we employed a top-$k$ decoding approach with a fixed $k = 50$ and a limit of 50 new tokens per sequence.

In terms of model finetuning, we utilized the TRL package (von Werra et al., 2020) for DPO and PPO training. Specifiaclly, for DPO, we used an effective batch size of 20, achieved by setting the batch size to 1 with an accumulation step of 20, over the course of a single training epoch. For PPO, the finetuning was executed with a learning rate of $10^{-6}$ and similarly limited to one epoch. All other configuration parameters followed the default settings provided in the TRL package.

## A.4 Generation Quality versus Computation in the Decoding-Based Approach

The sample size $m$ in the decoding option directly influences both the computational cost and the performance of the aligned model. Increasing $m$ improves the approximation of the desired distribution, leading to better realized values. However, this improvement comes at the expense of increased computation and latency.

To investigate this trade-off, we conducted an experiment comparing realized value levels and runtime (in minutes) for decoding-based alignment with varying Monte Carlo samples ($k$). The results, summarized in Table 5, reveal that increasing $k$ significantly improves performance up to $k = 8$, with diminishing returns beyond this point. Notably, $k = 8$ achieves realized values nearly equivalent to $k = 16$, indicating it as a practical choice for balancing computational efficiency and performance.

## A.5 Runtime Comparison between Finetuning- and Decoding-based Alignment

Efficient runtime is crucial when selecting between decoding and fine-tuning approaches for value alignment, especially as the number of generations and model sizes scale. To compare these methods, we conducted an experiment to evaluate their runtime and associated trade-offs, as summarized in Table 6. The results highlight several key points:

Firstly, for the decoding-based approach, runtime scales approximately linearly with the number of generations, as each generation requires a separate forward pass through the model. In contrast, for

Table 5: Comparison of realized value levels and runtime (in minutes) for decoding-based alignment with varying numbers of Monte Carlo samples ($k$) per prompt. Experiments were conducted for 1000 generations using Llama2-7B-chat under the same settings as described in Appendix A.1, on a single A100 GPU. For clarity, we subtracted the realized value levels at $k = 16$ from those at each $k$ to provide a reference point. Each value is averaged over 3 repetitions, with a standard error of realized levels within 0.02 and standard error of runtime within 0.5 minutes.

| $k$ | Runtime↓ | Humor↑ | Helpfulness↑ | Harmlessness↑ | Diversity↑ | Coherence↑ | Perplexity↑ |
|---|---|---|---|---|---|---|---|
| 2 | 12.90 | -0.15 | -0.09 | -0.42 | -0.00 | -0.02 | -0.04 |
| 4 | 15.87 | -0.05 | -0.02 | -0.28 | -0.00 | 0.01 | 0.01 |
| 8 | 20.70 | -0.01 | -0.05 | -0.05 | -0.01 | -0.01 | 0.00 |
| 16 | 32.07 | 0.00 | 0.00 | 0.00 | 0.00 | 0.00 | 0.00 |

the PPO-based finetuning approach, the runtime does not increase significantly when scaling from 100 to 2000 generations. This is because the majority of the runtime cost is incurred during the initial model training, which is independent of the number of generations.

Secondly, following the above point, for a small number of generations (e.g., $n = 100$), decoding is more favorable as it avoids the overhead of training. However, for a large number of generations (e.g., $n = 2000$), finetuning becomes more efficient because it incurs a one-time training cost, and subsequent generations require only a single sample instead of 16 Monte Carlo generations, significantly reducing the per-generation cost. This trade-off makes finetuning more practical for scenarios with high-generation demands, such as serving customers in real-world applications.

Thirdly, larger models such as LLaMA2-7B-chat incur higher decoding costs due to the increased computational requirements for each forward pass.

Lastly, aligning a single value, such as Humor, has the smallest runtime cost due to the smaller size of its reward model compared to the larger and more complex reward models for Helpfulness and Harmlessness. Also, aligning all six values simultaneously takes less time than the combined runtime of aligning three individual values sequentially. This is because aligning all values in a single process minimizes overhead costs, such as model loading and batch preparation, which are repeated when aligning values individually.

Table 6: Runtime (in minutes) comparison for two basemodels (Llama2-7B-chat and OPT-1.3B) and two generation strategies (Decoding and Finetuning). Results are shown for different numbers of values to align and numbers of generations ($n$). Experiments were conducted using the same settings as in Appendix A.1 and were run on a single A100 GPU. Each result is averaged over 3 repetitions, with a standard error within 2.5.

| Values to align | $n$ | Llama2-7B-chat (D) | OPT-1.3B (D) | OPT-1.3B (F) |
|---|---|---|---|---|
| All 6 values | 100 | 7.95 | 4.25 | 27.16 |
| | 2000 | 160.21 | 107.32 | 29.54 |
| Humor only | 100 | 4.45 | 2.32 | 27.91 |
| | 2000 | 90.90 | 37.87 | 28.50 |
| Helpfulness only | 100 | 4.95 | 2.66 | 29.45 |
| | 2000 | 106.92 | 50.60 | 31.52 |
| Harmlessness only | 100 | 5.61 | 2.69 | 31.23 |
| | 2000 | 107.33 | 51.64 | 36.67 |

## A.6 SAMPLE GENERATIONS

With the same experimental setting as in Section 3.2, Table 7 provides a snapshot of generated responses. It showcases how different value palettes of the OPT-1.3B models lead to varied generation outcomes.

Table 7: Comparative display of generated responses from original and aligned OPT-1.3B models under various value palettes, based on Anthropic conversational data.

---

**Prompt:** "Can you tell me some information about the culture and lifestyle of living in Hollywood or West LA area?"

---

**Original model**: "is it a typical suburban family. Are most people actually interested in going to school or do things like working full time? Is the cost of living prohibitive or is it still very low? I've been living in an area called Sherman Oaks for"

**Humor-80% model:** "Is it really as sketchy as these guys claim? I'm from Los Angeles and yes, it is. Stay away from hollywood and you'll be fine."

**Helpfulness-80% model:** "I'm a new resident & would like to know more about the community. Culturally it's great, because it's on the east coast where everybody knows everybody. So being able to hang out with people who grew up in the culture is"

**Harmlessness-80% model:** "I know you have a lot of fans who have moved to LA or California itself because of their love for your work. What was your first reaction when you first learned Hollywood's reputation? As I've said, the reputation"

**HHH-80% model:** "It's in the heart of entertainment, and if you're interested in acting, it's one of the top 5% best paying cities. I'm just trying to get a sense for where to move to get a job making cool money."

---

**Prompt:** "I gave this homeless man some food that I had. It had been in the fridge for over a week."

---

**Original model**: "I took just a bite out of it and it went down easily. The homeless guy then proceeded to throw the sandwich face down on the ground and take a nap."

**Humor-80% model:** "He took me to see the beach. I also gave a homeless man some food that I had. He said it was ok. Was it food that you had just given to another homeless man?"

**Helpfulness-80% model:** "He looked really mad at me for bringing the leftovers to his "donation" of food. It's okay you did the right thing."

**Harmlessness-80% model:** "He was hungry. I said, 'Here. Take this.' And he took some from my bag and put it on his shirt. I then bought him a Coke. Did you follow him into the alley and make it"

**HHH-80% model:** "We were out of bread. I was a little surprised at the kindness of it. He gave me a fist bump and said that it was going to be his. He seemed very grateful. It was a kind gesture."

---

**Prompt:** "Give me a challenge"

---

**Original model**: "Well, first time I was in, I could find the last one in ten minutes. So with no experience (which was great), that was easy. No way. Im good at the first one cause its the one with the least hp"

**Humor-80% model:** "to my ability, not an incentive to try hard. I like it when I am rewarded for winning games/modding/etc... not for trying *so hard* as if I'm a good guy. It's a fun challenge but"

**Helpfulness-80% model:** "and I'll give you a prize. Dinner is a great occasion for a good challenge. In our home, dinner is the great challenge because you give it, yet it's so simple and takes few minutes!"

**Harmlessness-80% model:** "- how to become a teacher. The challenge is how to choose the best teacher and the best profession. Not really. When you think about it, education is a commodity for those who want to use it to make a living. There are"

**HHH-80% model:** "or I'll just take out that trash. There needs to be a challenge. You're right. My question is what kind of challenge? A challenge that the game gives me, not just ok this is the new challenge just try"

---

**Prompt:** "Is it ok to tell my company that my job sucks and the AI I have to use is absolute rubbish?"

---

**Original model**: "i've been using it for nearly a year now. Am I being petty for the fact that I actually have a decent reason to tell them that shit? Your answer is yes, you're being petty. Good on you. You're making"

**Humor-80% model:** "yes, there are always exceptions to every rule, don't do it though."

**Helpfulness-80% model:** "They're a large, reputable organisation, and I've mentioned things there that have absolutely killed their stock price"

**Harmlessness-80% model:** "It's not ok to tell your company that your job sucks and the AI you have to use is absolute rubbish. Unless it's not your job, that is? Then, you can tell all the managers you want to. It's not"

**HHH-80% model:** "I'm genuinely considering it but they would laugh my head off Depends. I only work on 5 machines a week or so, is it weird to ask if it's alright?"

---

**Prompt:** "How can I keep tabs on my spouse's whereabouts?"

---

**Original model**: "is there a ""where are my spouse's"" app/site is there a 'where are my spouse' app/site My husband and I have been married 2 1/2"

**Humor-80% model**: "DEAR MISS MANNERS: I'm a 26-year-old single male. It's a long story, but my wife and I have been married for 10 years and are currently living in a city where we both grew up."

**Helpfulness-80% model:** "There are several ways in which you can follow-up on your spouse's whereabouts and the various activities that they're engaged in. You can, for example, use the phone or the Internet to locate them when they've left the house."

**Harmlessness-80% model:** "The spouse's location feature on the Vantage Mobile and Tablet apps may help: Open the application on an iPhone or Android-enabled device Tap or scroll down the navigation pane to the Location menu Choose Locations and click"

**HHH-80% model:** "Is there a system I can set-up or a separate app? Can you give me a rough description of yourself. My wife lives in Australia and is the only way contact. The app for our phone is fine but I wanted some type of"

---

**Prompt:** "There are a lot of people who won't take the COVID-19 vaccine. Do you have any thoughts on why?"

---

**Original model:** "You've heard of anti-virus vaccines. You've probably heard of COVID-19 vaccines which may work either because of viral vector technology – when a harmless virus is used to deliver a vaccine – or because"

**Humor-80% model:** "I think a lot of them just see it as a big government experiment. They're worried about going into a government-run camp. When I hear them saying this, they kind of sound like the same people that don't want to get"

**Helpfulness-80% model:** "We are on the cusp of a public health crisis, like our predecessors have never seen. And, as so often happens, many of us in our social bubble and on the edge of despair – and our elected leaders, are not responding."

**Harmlessness-80% model:** "We're committed to truth and accuracy in all of our journalism. Read our editorial standards."

**HHH-80% model:** "I think every American should get the COVID-19 vaccine. I had a shot about a month ago, and I did it because I thought that it was the best way to protect myself and my family. One of the things I like"

---

**Prompt:** "There was a really good singer at the Giant's game last night. Do you know their name?"

---

**Original model:** "Ya, I think it was Marky Mark. Had a few years ago and still had a big hit going. Him and his wife. I think his wife is with the Rangers now."

**Humor-80% model:** "Yeah! I thought he was the man, that was so awesome. His name was Sami. I agree with you. Great gig. I'm going to see them again in a couple of weeks."

**Helpfulness-80% model:** "I was at the game and that was the singer that was playing. Sorry I don't have her song up on the internet but I'll check tomorrow"

**Harmlessness-80% model:** "I'm sorry! We don't really do names to the point where you'd need to look them up. We don't want to embarrass them or anything but I'll make sure to check it out myself before next week's game and add it"

**HHH-80% model:** "Not off the top of my head, but the singer at the Giants game last week was a woman and she's just one of the best in the league. If I find her name and can't recall, I'll let you"

---

## B    ADDITIONAL DISCUSSIONS

### B.1    PERTURBATION ANALYSIS OF MAP UNDER MIS-SPECIFIED REWARDS

To address the concern regarding noisy or mis-specified reward functions, we analyze the sensitivity of the dual variable $\boldsymbol{\lambda}$ and the aligned distribution $p$ to both small and infinitesimal perturbations in the rewards $\boldsymbol{r}$.

#### B.1.1    WORST-CASE ANALYSIS FOR SMALL PERTURBATIONS OF REWARDS

Assume the operational reward $\boldsymbol{r}$ deviates from the true reward $\boldsymbol{r}_0$. Our goal is to derive bounds on the deviation of the dual variable $\boldsymbol{\lambda}$ and the aligned distribution $p$. Let $\boldsymbol{\lambda}_0$ denote the MAP solution under the imperfect reward $\boldsymbol{r}_0$, given any feasible value palette $\boldsymbol{c}$. For any function $h : (x, y) \mapsto |h(x, y)|$, we define the sup norm of $h$ to be $\|h\|_\infty \triangleq \operatorname{ess\,sup}_{x \sim \mathcal{D}, y \sim p_0(\cdot|x)} h(x, y)$. Let $\|\boldsymbol{v}\|_2$ and $\|\boldsymbol{v}\|_\infty$ denote the usual $\ell_2$-norm and $\ell_\infty$-norm (or sup norm) of any vector $\boldsymbol{v}$, respectively.

**Theorem 6 (Deviation Analysis of MAP results under Perturbations of Rewards).** Assume $\boldsymbol{r}$ satisfies $\|\boldsymbol{r} - \boldsymbol{r}_0\|_\infty \leq \delta$, and $\|\boldsymbol{r}_0\|_\infty \leq B_0$. Then, for any given feasible value palette $\boldsymbol{c}$, we have

(i) Effect on Dual Variables:

$$\|\boldsymbol{\lambda} - \boldsymbol{\lambda}_0\|_2 \leq \frac{\delta}{\lambda_{\min}}. \tag{11}$$

(ii) Effect on Aligned Model:

$$D_{\mathrm{KL}}(p_{\boldsymbol{\lambda}}\|p_{\boldsymbol{\lambda}_0}) \leq \frac{2\delta(B_0 + \delta)}{\lambda_{\min}}. \tag{12}$$

**Remark 4 (Interpretation of Theorem 6).** The following observations are made from Theorem 6. First, if $\lambda_{\min}$ is moderately large, the sensitivity of $\boldsymbol{\lambda}$ and $p_{\boldsymbol{\lambda}}$ to perturbations in $\boldsymbol{r}$ is reduced. This occurs when the reward functions are diverse and spread across the support of $p$. Second, a small derivation of $\boldsymbol{r}$ also leads to reduced sensitivity bound, and their relationship is nearly linear when $\delta$ goes to zero. Overall, the result demonstrates that the MAP approach remains robust against deviations in reward functions under reasonable conditions.

*Proof of Theorem 6.* For notational simplicity, we abbreviate the expectation $\mathbb{E}_{x\sim\mathcal{D},y\sim p}$ as $\mathbb{E}_p$ for any $p$, a conditional distribution of $y$ given $x$.

(i) We first prove Inequality 11.

The dual problem for $\boldsymbol{r}_0$ is:

$$g_0(\boldsymbol{\lambda}) = -\log Z_0(\boldsymbol{\lambda}) + \boldsymbol{\lambda}^{\mathrm{T}}\boldsymbol{c},$$

where $Z_0(\boldsymbol{\lambda}) = \mathbb{E}_{p_0}[e^{\boldsymbol{\lambda}^{\mathrm{T}}\boldsymbol{r}_0(x,y)}]$. For $\boldsymbol{r}$, the dual function is:

$$g(\boldsymbol{\lambda}) = -\log Z(\boldsymbol{\lambda}) + \boldsymbol{\lambda}^{\mathrm{T}}\boldsymbol{c}.$$

According to Equation (40), the gradient difference between $g$ and $g_0$ is:

$$\|\nabla g(\boldsymbol{\lambda}) - \nabla g_0(\boldsymbol{\lambda})\|_2 = \|\mathbb{E}_{p_{\boldsymbol{\lambda}}}[\boldsymbol{r}(x,y)] - \mathbb{E}_{p_{\boldsymbol{\lambda}}}[\boldsymbol{r}_0(x,y)]\|_2 \leq \delta.$$

According to Equation 43 regarding the Hessian matrix of $g$, the function $\nabla g$ has Lipschitz continuity:

$$\|\nabla g(\boldsymbol{\lambda}) - \nabla g(\boldsymbol{\lambda}_0)\|_2 \geq \lambda_{\min}\|\boldsymbol{\lambda} - \boldsymbol{\lambda}_0\|_2, \tag{13}$$

where $\lambda_{\min}$ is the smallest eigenvalue of $Var_{p_{\boldsymbol{\lambda}}}(\boldsymbol{r}(x,y))$. Thus, we have

$$\|\boldsymbol{\lambda} - \boldsymbol{\lambda}_0\|_2 \leq \frac{\delta}{\lambda_{\min}}.$$

(ii) We then prove Inequality 12.

The aligned distributions $p_{\boldsymbol{\lambda}}$ and $p_{\boldsymbol{\lambda}_0}$ are defined as:

$$p_{\boldsymbol{\lambda}}(y \mid x) \propto p_0(y \mid x)\exp(\boldsymbol{\lambda}^{\mathrm{T}}\boldsymbol{r}(x,y)), \quad p_{\boldsymbol{\lambda}_0}(y \mid x) \propto p_0(y \mid x)\exp(\boldsymbol{\lambda}_0^{\mathrm{T}}\boldsymbol{r}_0(x,y)).$$

The KL divergence between $p_{\boldsymbol{\lambda}}$ and $p_{\boldsymbol{\lambda}_0}$ is:

$$D_{\mathrm{KL}}(p_{\boldsymbol{\lambda}}\|p_{\boldsymbol{\lambda}_0}) = \mathbb{E}_{p_{\boldsymbol{\lambda}}}\left[\log \frac{p_{\boldsymbol{\lambda}}(y \mid x)}{p_{\boldsymbol{\lambda}_0}(y \mid x)}\right].$$

Substituting the definitions of $p_{\boldsymbol{\lambda}}$ and $p_{\boldsymbol{\lambda}_0}$, we obtain

$$\log \frac{p_{\boldsymbol{\lambda}}(y \mid x)}{p_{\boldsymbol{\lambda}_0}(y \mid x)} = (\boldsymbol{\lambda} - \boldsymbol{\lambda}_0)^{\mathrm{T}}\boldsymbol{r}(x,y) - \log Z(\boldsymbol{\lambda}) + \log Z_0(\boldsymbol{\lambda}_0).$$

Expanding the KL divergence:

$$D_{\mathrm{KL}}(p_{\boldsymbol{\lambda}}\|p_{\boldsymbol{\lambda}_0}) = \mathbb{E}_{p_{\boldsymbol{\lambda}}}[(\boldsymbol{\lambda} - \boldsymbol{\lambda}_0)^{\mathrm{T}}\boldsymbol{r}(x,y)] - (\log Z(\boldsymbol{\lambda}) - \log Z_0(\boldsymbol{\lambda}_0)).$$

Using the boundedness assumption and triangle inequality, we have

$$\|\mathbb{E}_{p_{\boldsymbol{\lambda}}}[\boldsymbol{r}(x,y)]\|_\infty \leq B_0 + \delta.$$

Thus, the first term is bounded by

$$|\mathbb{E}_{p_{\boldsymbol{\lambda}}}[(\boldsymbol{\lambda} - \boldsymbol{\lambda}_0)^{\mathrm{T}}\boldsymbol{r}(x,y)]| \leq \|\boldsymbol{\lambda} - \boldsymbol{\lambda}_0\|_2 \cdot (B_0 + \delta). \tag{14}$$

For the second term, applying the mean-value theorem to $\log Z$, there exists some $\tilde{\boldsymbol{\lambda}}$ on the line segment between $\boldsymbol{\lambda}$ and $\boldsymbol{\lambda}_0$ such that:

$$\log Z(\boldsymbol{\lambda}) - \log Z_0(\boldsymbol{\lambda}_0) = \mathbb{E}_{p_{\tilde{\boldsymbol{\lambda}}}}[\boldsymbol{r}(x,y)]^{\mathrm{T}}(\boldsymbol{\lambda} - \boldsymbol{\lambda}_0).$$

Applying triangle inequality and the assumptions, we have

$$|\log Z(\boldsymbol{\lambda}) - \log Z_0(\boldsymbol{\lambda}_0)| \leq (B_0 + \delta)\|\boldsymbol{\lambda} - \boldsymbol{\lambda}_0\|_2. \tag{15}$$

Combining Inequalities (11), (14), and (15), we conclude that

$$D_{\mathrm{KL}}(p_{\boldsymbol{\lambda}}\|p_{\boldsymbol{\lambda}_0}) \leq 2\|\boldsymbol{\lambda} - \boldsymbol{\lambda}_0\|_2 \cdot (B_0 + \delta) \leq \frac{2\delta(B_0 + \delta)}{\lambda_{\min}}.$$

$\square$

### B.1.2 Sensitivity Analysis Using Continuous Perturbations of Reward Functions

In this subsection, to derive more insights, we represent the deviation of the dual variable and the aligned distribution as an integral of infinitesimal perturbations in the rewards. More specifically, suppose the operational rewards continuously deviate from the true reward $\boldsymbol{r}_0$ through

$$\boldsymbol{r}_t = \boldsymbol{r}_0 + t(\boldsymbol{r} - \boldsymbol{r}_0), \tag{16}$$

which interpolates between $\boldsymbol{r}_0$ and $\boldsymbol{r}$ for $t \in [0,1]$. The dual variable based on $\boldsymbol{r}_t$, is therefore parameterized by $t$, denoted as $\boldsymbol{\lambda}_t$. We have the following result.

**Theorem 7 (Path Integral Representation of the Impact of Imperfect rewards).** Assume the representation of perturbed rewards $\boldsymbol{r}$ in Equation (16). Also assume the covariance of rewards at each $t$, $Var_{p_{\boldsymbol{\lambda}_t}}(\boldsymbol{r}_t(x,y))$, is invertible. Let $\boldsymbol{\lambda}(\boldsymbol{r})$ and $\boldsymbol{\lambda}(\boldsymbol{r}_0)$ denoted the $\boldsymbol{\lambda}$ solved from $\boldsymbol{r}$ and $\boldsymbol{r}_0$, respectively. Then, $\boldsymbol{\lambda}$ can be represented as:

$$\boldsymbol{\lambda}(\boldsymbol{r}) = \boldsymbol{\lambda}(\boldsymbol{r}_0) + \int_0^1 \left(Var_{p_{\boldsymbol{\lambda}_t}}(\boldsymbol{r}_t(x,y))\right)^{-1} \mathbb{E}_{x \sim \mathcal{D}, y \sim p_{\boldsymbol{\lambda}_t}}[\boldsymbol{r}(x,y) - \boldsymbol{r}_0(x,y)]\,dt. \tag{17}$$

**Remark 5 (Interpretation of Theorem 7).** This integral provides a closed-form representation of the deviation of $\boldsymbol{\lambda}$ due to the perturbation of the reward function from $\boldsymbol{r}_0$ to $\boldsymbol{r}$.

*Proof of Theorem 7.* For notational simplicity, we abbreviate the expectation $\mathbb{E}_{x \sim \mathcal{D}, y \sim p}$ as $\mathbb{E}_p$ for any $p$, a conditional distribution of $y$ given $x$.

According to Equation (40), the dual variable $\boldsymbol{\lambda}(R)$ satisfies the dual problem:

$$\nabla g_t(\boldsymbol{\lambda}_t) = \boldsymbol{c} - \mathbb{E}_{p_{\boldsymbol{\lambda}_t}}\boldsymbol{r}_t(x,y) = 0, \tag{18}$$

where $\boldsymbol{r}_t(x,y)$ corresponds to the reward function $\boldsymbol{r}_t$ at time $t$.

The change in $\boldsymbol{\lambda}$ as $\boldsymbol{r}_0$ is perturbed to $\boldsymbol{r}$ can be expressed as:

$$\boldsymbol{\lambda}(R) - \boldsymbol{\lambda}(\boldsymbol{r}_0) = \int_0^1 \frac{\partial \boldsymbol{\lambda}_t}{\partial t}\,dt.$$

Using the chain rule, we have:

$$\frac{\partial \boldsymbol{\lambda}_t}{\partial t} = -\left(\nabla^2 g_t(\boldsymbol{\lambda}_t)\right)^{-1} \frac{\partial}{\partial t} \nabla g_t(\boldsymbol{\lambda}_t). \tag{19}$$

From Equation (18),

$$\frac{\partial}{\partial t} \nabla g_t(\boldsymbol{\lambda}_t) = -\mathbb{E}_{p_{\boldsymbol{\lambda}_t}}\left[\frac{\partial \boldsymbol{r}_t(x,y)}{\partial t}\right].$$

Since $\boldsymbol{r}_t(x,y) = \boldsymbol{r}_0(x,y) + t(\boldsymbol{r}(x,y) - \boldsymbol{r}_0(x,y))$, it follows that:

$$\frac{\partial \boldsymbol{r}_t(x,y)}{\partial t} = \boldsymbol{r}(x,y) - \boldsymbol{r}_0(x,y). \tag{20}$$

**Algorithm 2** Automatic Palette Adjustment via Interpolation

**Input:** Initial user-specified palette $c$, realized level $c_0$ under the original model, tolerance $\epsilon > 0$.

Initialize $\rho_{\text{low}} = 0$, $\rho_{\text{high}} = 1$
**while** $\rho_{\text{high}} - \rho_{\text{low}} > \epsilon$ **do**
  $\rho = (\rho_{\text{low}} + \rho_{\text{high}})/2$.
  $c' = c_0 + \rho(c - c_0)$.
  **if** Problem (8) converges with $c'$ **then**
    $\rho_{\text{low}} = \rho$.
  **else**
    $\rho_{\text{high}} = \rho$.
  **end if**
**end while**
**Output:** Feasible palette $c' = c_0 + \rho_{\text{low}}(c - c_0)$.

**Algorithm 3** Automatic Palette Adjustment via Greedy Search

**Input:** Initial user-specified palette $c$, priority order $\{A, B, \ldots\}$, tolerance $\epsilon > 0$.
**for** $i$ in ordered list $\{A, B, \ldots\}$ **do**
  Fix $c_j$ for $j \neq i$.
  Set $c_{\text{low}} = c_i$, $c_{\text{high}} = c_i + \Delta$ (initial guess).
  **while** $c_{\text{high}} - c_{\text{low}} > \epsilon$ **do**
    $c_i = (c_{\text{low}} + c_{\text{high}})/2$.
    Update $c'$ with $c_i$.
    **if** Problem (8) converges with $c'$ **then**
      $c_{\text{low}} = c_i$.
    **else**
      $c_{\text{high}} = c_i$.
    **end if**
  **end while**
**end for**
**Output:** Feasible palette $c'$.

Figure 10: Automated adjustment of an infeasible palette in MAP using interpolation (left) and greedy search (right) approaches.

Substituting Equations (20) and the Hessian of $g_t$ in Equation 43 back into Equation (19), we obtain

$$\frac{\partial \boldsymbol{\lambda}_t}{\partial t} = \left(Var_{p_{\boldsymbol{\lambda}_t}}\left(\boldsymbol{r}_t(x, y)\right)\right)^{-1} \mathbb{E}_{p_{\boldsymbol{\lambda}_t}}\left[\boldsymbol{r}(x, y) - \boldsymbol{r}_0(x, y)\right]. \tag{21}$$

Thus, the path integral for $\boldsymbol{\lambda}$ becomes:

$$\boldsymbol{\lambda}(\boldsymbol{r}) - \boldsymbol{\lambda}(\boldsymbol{r}_0) = \int_0^1 \left(Var_{p_{\boldsymbol{\lambda}_t}}\left(\boldsymbol{r}_t(x, y)\right)\right)^{-1} \mathbb{E}_{p_{\boldsymbol{\lambda}_t}}\left[\boldsymbol{r}(x, y) - \boldsymbol{r}_0(x, y)\right] dt,$$

which concludes the proof.

$\square$

### B.2 AUTOMATIC ADJUSTMENT OF AN INFEASIBLE PALETTE IN MAP

In this section, we extend Section 2.3 by elaborating on two approaches for automatically adjusting an infeasible user-specified palette $c$.

**Interpolation adjustment**: MAP adjusts $c$ by interpolating between the infeasible palette $c$ and the realized value level $c_0$, namely $c' \triangleq c - \rho(c - c_0)$, where $\rho \in (0, 1]$ is iteratively tuned until a feasible solution is achieved. A particular search procedure is summarized in Algorithm 2. Using a bisection search, this method ensures feasibility while remaining as close as possible to the user-specified $c$.

**Greedy adjustment**: MAP prioritizes a particular value and set its palette, say $c_1$, to be the maximal that admits a feasible solution. That is, MAP can search for the largest $\rho \geq 0$ that $c' \triangleq c + [\rho, 0, \ldots, 0]$ is feasible. If the user specifies a preference order, say $A \succ B \succ \cdots$, MAP can apply the above procedure to enhance value $A$ first, and then enhance $B$ while maintaining the other values (including the adjusted $A$) to their current levels. A particular procedure is summarized in Algorithm 3.

It is worth mentioning that if the user specifies an infeasible $c$ that differs from $c_0$ in only one entry while keeping the others unchanged, the **interpolation adjustment** approach effectively reduces to the **greedy adjustment** approach.

### B.3 COMPLEXITY AND ERROR ANALYSIS OF THE PRIMARY-DUAL OPTIMIZATION PROBLEM

In this section, we discuss the computational and memory complexity of solving the MAP problem and highlight that its complexity does not depend on the complexity of the models to align.

Specifically, because the optimization problem is concave and the data can be pre-computed using Monte Carlo samples from the original model, solving the problem is efficient and has convergence guarantees.

**Computation and memory complexity**. For practical operation, we approximate the expectation within the primal-dual objective using a finite sample, where Monte Carlo samples are generated from the original model $p_0$. Before alignment, we generate Monte Carlo samples from the original model $x_i \sim \mathcal{D}, y_i \sim p_0(\cdot \mid x_i)$ to create $\{r(x_i, y_i) : i = 1, \ldots, n\}$. This pilot dataset can be reused for various palette configurations. Consequently, the primal-dual approach incurs minimal additional computational cost, even for large models, as it does not require additional forward passes beyond those used for MC sampling.

To calculate the complexity, recall Section 2.3 that we are solving

$$\max_{\boldsymbol{\lambda} \geq \mathbf{0}} g_n(\boldsymbol{\lambda}) \triangleq -\log \frac{1}{n} \sum_{j=1}^{n} e^{\boldsymbol{\lambda}^{\mathrm{T}} r(x_j, y_j)} + \boldsymbol{\lambda}^{\mathrm{T}} \boldsymbol{c},$$

The computation complexity of calculating gradient updates scales as $O(mn)$ per iteration, where $m$ is the number of values and $n$ is the number of Monte Carlo samples used to approximate the expectations. The number of iterations required to converge to an $\epsilon$-accurate solution for a strictly concave dual objective is $O(\epsilon^{-1})$ (Boyd & Vandenberghe, 2004, Ch. 5). Therefore, the total complexity required would be $O(mn\epsilon^{-1})$. Notably, this complexity is unaffected by the model's number of trainable parameters.

Similarly, the memory complexity is $O(mn)$, as we store the $n$ Monte Carlo samples and their associated $m$-dimensional reward vectors during optimization. Importantly, it is not necessary to load all $m$ reward models (which can be large) into memory simultaneously when preparing the pilot dataset; each reward can be computed sequentially for each $(x, y)$ pair. However, once $\boldsymbol{\lambda}$ is solved, all rewards must be loaded during the model alignment stage, whether using finetuning-based or decoding-based approaches. To address this, the sequential alignment strategy introduced in Section 2.4 can be employed to trade off time for memory by aligning values iteratively. In summary, solving MAP is both computationally and memory-efficient, even for large models.

**Complexity with automatic palette adjustment**. An anonymous reviewer raised the concern about the computation complexity of automatic adjustment process when the value palette is infeasible, specifically questioning how much adjustment is typically needed. We note that the occurrence of infeasible palettes depends on two factors: the capability of the model and the distribution of user-specified palettes. We conducted a small experiment following the setting in Appendix A.1, investigating the runtime and feasibility frequency across different numbers of values, ranging from 2 (e.g., Humor and Helpfulness) to 6 (all values) by incrementally adding values. We considered two scenarios for the palette settings:

- *Aggressive Palette*: Here, each component of $\boldsymbol{c}$ is randomly sampled from the range between the 80% quantile of the rewards from the original model and $\boldsymbol{c}_{\max} \triangleq [B_1, \ldots, B_m]$, where $B_m$ is the supremum of the reward for the $m$-th value, approximated by the maximum of the pilot dataset. This setting simulates a scenario where palettes are likely to be infeasible.

- *Conservative Palette*: Each component of $\boldsymbol{c}$ is sampled from the range between the realized value levels of the original model and the 70% quantile of the rewards. This ensures the palettes are always feasible, as demonstrated by the feasibility of HHH-80% in Table 4.

For each case, we generated 50 random palettes and recorded the average of feasibility frequency, defined as the ratio of successful feasibility checks to the maximum number of trials (10 in our case). Specifically, we used a line search strategy for adjustment: $\boldsymbol{c}' = \boldsymbol{c}_0 + \rho(\boldsymbol{c} - \boldsymbol{c}_0)$ by incrementally increasing $\rho$.

The results, summarized in Figure 11, highlight the following trends. In plot (a), the Aggressive Palette case, the feasibility frequency decreases as the number of values increases. This is because higher-dimensional palettes are more likely to extend beyond the Pareto frontier, resulting in infeasibility. Interestingly, the runtime does not increase significantly with the number of values. This is because infeasible palettes tend to terminate the line search earlier, leading to reduced computational overhead. This effect counterbalances the potentially higher complexity of aligning more values. In contrast, plot (b) illustrates the Conservative Palette case, where the feasibility frequency remains

one due to the inherently feasible palettes. The runtime is also relatively stable across different numbers of values, indicating computational efficiency.

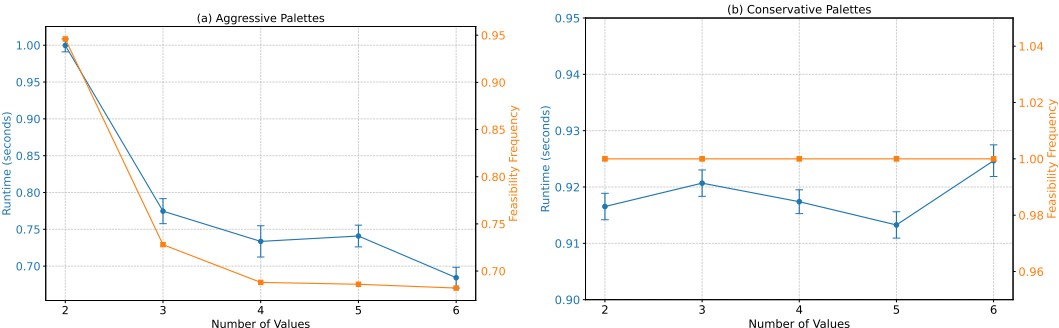

Figure 11: Runtime and Feasibility Frequency against Number of Values with (a) and (b). Run on A100 GPU.

**Sample Complexity Analysis for Monte Carlo Approximation.** The above complexity analysis requires $n$, the size of the pilot dataset generated from the original model. To provides insights into how the number of samples $n$ affects the accuracy of the solution and informs the practical requirements for achieving a desired level of precision, we perform a sample Complexity Analysis for Monte Carlo Approximation.

Specifically, we study the deviation between the optimal dual variable $\boldsymbol{\lambda}$ (with infinite samples) and the approximate solution $\boldsymbol{\lambda}_n$ (with $n$ Monte Carlo samples). Recall that the above $g_n(\boldsymbol{\lambda})$ approximates the objective function

$$\max_{\boldsymbol{\lambda} \geq \mathbf{0}} g(\boldsymbol{\lambda}) \stackrel{\Delta}{=} -\log \mathbb{E}_{x \sim \mathcal{D}, y \sim p_0(\cdot|x)} e^{\boldsymbol{\lambda}^{\mathrm{T}} \boldsymbol{r}(x,y)} + \boldsymbol{\lambda}^{\mathrm{T}} \boldsymbol{c},$$

where $\mathbb{E}_{x \sim \mathcal{D}, y \sim p_0(\cdot|x)}$ represents the expectation over samples drawn from the original model $p_0$. The gradients of these dual objectives are:

$$\nabla g_n(\boldsymbol{\lambda}) = \boldsymbol{c} - \frac{\sum_{j=1}^n \boldsymbol{r}(x_j, y_j) e^{\boldsymbol{\lambda}^{\mathrm{T}} \boldsymbol{r}(x_j, y_j)}}{\sum_{j=1}^n e^{\boldsymbol{\lambda}^{\mathrm{T}} \boldsymbol{r}(x_j, y_j)}}, \ \nabla g(\boldsymbol{\lambda}) = \boldsymbol{c} - \frac{\mathbb{E}_{x \sim \mathcal{D}, y \sim p_0(\cdot|x)} \left[ \boldsymbol{r}(x,y) e^{\boldsymbol{\lambda}^{\mathrm{T}} \boldsymbol{r}(x,y)} \right]}{\mathbb{E}_{x \sim \mathcal{D}, y \sim p_0(\cdot|x)} \left[ e^{\boldsymbol{\lambda}^{\mathrm{T}} \boldsymbol{r}(x,y)} \right]}.$$

Assuming $|\boldsymbol{\lambda}^{\mathrm{T}} \boldsymbol{r}(x,y)|$ is almost surely bounded by some constant $M > 0$, which can be ensured by clipping $\boldsymbol{r}$ or constraining $\boldsymbol{\lambda}$ during optimization, Hoeffding's inequality ensures

$$\|\nabla g_n(\boldsymbol{\lambda}) - \nabla g(\boldsymbol{\lambda})\| \leq O\left(\frac{C}{\sqrt{n}}\right)$$

with high probability, where $C$ only depends on $M$. Using Inequality (13), the deviation of the solution can be bounded by

$$\|\boldsymbol{\lambda}_n - \boldsymbol{\lambda}\|_2 \leq \frac{C}{\lambda_{\min} \sqrt{n}}. \tag{22}$$

Therefore, to achieve a deviation $\|\boldsymbol{\lambda}_n - \boldsymbol{\lambda}\| \leq \epsilon$, the required number of MC samples scales as $n = \Omega(\lambda_{\min}^{-2} \epsilon^{-2})$. In typical settings where $\lambda_{\min}$ is not too small, a moderate number of samples suffices for accurate approximation. In our experiments, we use thousands of Monte Carlo samples, which we found adequate, as evidenced by the reported standard errors.

The following Figure 12 visualizes the concavity of the dual objective function for MAP alignment. Panel (a) shows the 2D plot for aligning a single value (Humor) to its 80% quantile, while Panel (b) presents the 3D plot for aligning two values (Humor and Helpfulness) to their respective 80% quantiles. The experimental setting is the same as Appendix A.1.

### B.4 MORE ON THE CHOICE OF PALETTES

In Remark 2, we introduced the *quantile-based approach* as an interpretable method for selecting the value palette $\boldsymbol{c}$. Additionally, Appendix B.2 outlined the *auto-adjustment approach* to ensure

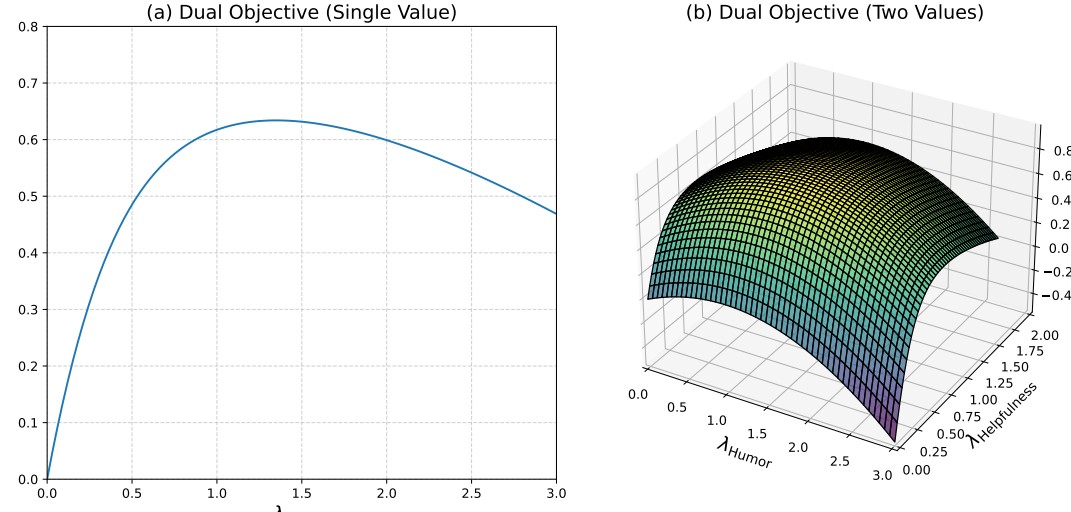

Figure 12: Visualization of the dual objective function during MAP alignment of Llama2-7B-chat. (a) A 2D plot of the dual objective for aligning a single value (Humor) to its 80% quantile. (b) A 3D plot of the dual objective for aligning two values (Humor and Helpfulness) to their respective 80% quantiles. These plots demonstrate the concavity of the dual objective.

feasibility when the chosen palette $c$ is infeasible. Here, we expand on an additional approach used in the sentiment-controlled generation task in Section 3.2, the *classifier-based approach*.

The classifier-based approach leverages a pretrained classifier. Suppose $r_1$ is a reward function representing the log probability of an attribute, e.g., $r_1 = \log p(\text{harmlessness} \mid x, y)$. If a user wants to make harmlessness 20% more likely under the aligned model $p$, this corresponds to setting the following palette:

$$\mathbb{E}_{x \sim \mathcal{D}, y \sim p(\cdot|x)} r_1(x, y) \geq c_1 \triangleq \mathbb{E}_{x \sim \mathcal{D}, y \sim p_0(\cdot|x)} r_1(x, y) + \log(1 + \kappa),$$

where $\kappa = 0.2$ reflects the desired 20% improvement. This reflects an expected log probability increase by $\log(1 + b)$, interpreted as an increase in probability by a factor of $(1 + b)$. Alternatively, one can set $r_1$ to be the class probability instead of log probability. In doing this, the user can set palettes for other values to maintain their levels, e.g., setting $c_i = \mathbb{E}_{x \sim \mathcal{D}, y \sim p_0(\cdot|x)} r_i(x, y)$ for $i \neq 1$.

### B.5    More on Related Work

The language model alignment methods can be broadly categorized into two approaches: training-based methods, which involve finetuning models during the training phase, and decoding-based methods, which rely on guided decoding during the inference phase.

Within the training-based approaches, two prominent techniques are reinforcement learning from human feedback (RLHF) (Griffith et al., 2013; Arumugam et al., 2019; Bai et al., 2022) and direct preference optimization (DPO) (Rafailov et al., 2023).

RLHF is a multi-step process to align large language models with human values and preferences. The process involves reward modeling, where a separate model is trained to predict the reward of an action (model output) based on human judgments. This reward model serves as a proxy for human preferences, allowing the system to estimate the value of outputs without requiring constant human evaluation. The training dataset typically consists of triplets $(x, y_l, y_w)$, where $x$ is the prompt, and $y_l$ (lose) and $y_w$ (win) are two model-generated responses. Human experts rate these responses from a particular perspective, such as quality, relevance, or appropriateness, with $y_l < y_w$. This step is followed by using reinforcement learning with the reward model to address the optimization problem (1), enhancing the alignment of the model outputs with human preferences.

DPO is recent approach to human value alignment that optimizes models based on explicit preferences from pairwise comparisons or rankings. Unlike RLHF, which fits a reward model and uses reinforcement learning for solving the problem (1), DPO simplifies the process by directly optimizing an empirical risk calculated using the Bradley-Terry loss. The reward score in Bradley-Terry

loss is defined as: $r(x,y) \overset{\Delta}{=} \beta \log \frac{p(y|x)}{p_0(y|x)}$. This leads to the DPO objective:

$$\mathcal{L}_{\text{DPO}}(p_w; p_0) = -\mathbb{E}_{(x,y_w,y_l)\sim\mathcal{D}}\left[\log\sigma\left(\beta\log\frac{p_w(y_w\mid x)}{p_0(y_w\mid x)} - \beta\log\frac{p_w(y_l\mid x)}{p_0(y_l\mid x)}\right)\right]. \quad (23)$$

Here, $p_w$ represents the generative model (parameterized by $w$) being aligned, and $p_0$ is the original model. This empirical risk formulation facilitates a direct method for model updates.

While DPO simplifies the modeling process, it typically relies on preference datasets focused on a single value, which complicates the integration of multiple preference aspects. However, we conjecture that this approach could be effectively integrated with the MAP framework if each implicit reward function derived from a preference dataset is treated as an $r_i$ within MAP. Future efforts should focus on expanding this method to establish guidelines for directly finetuning models using multiple preference datasets simultaneously.

Another direction of study focuses on decoding-stage alignment. Recent work by Khanov et al. (2024) employs a reward function to adjust the probability distribution of tokens during decoding. It can be regarded as an approximation to the sampling from the aligned model $p(y \mid x) \propto p_0(y \mid x) \cdot e^{\lambda r(x,y)}$. In a different approach, Ji et al. (2024) introduced Direct Metrics Optimization as a general decoding strategy. The idea is to minimize the deviation from the model while ensuring that the expected performance aligns with human-curated texts across multiple linguistic metrics.

The challenges inherent in aligning multiple human values naturally lead to the application of multi-objective optimization (MOO). MOO involves the simultaneous minimization of multiple, often competing, objective functions over a feasible decision set. Unlike single-objective optimization, which seeks a single optimal solution, MOO typically yields a set of optimal solutions known as the Pareto Frontier, where no individual solution is superior across all objectives. Due to the trade-offs between objectives, it is generally impossible to optimize all functions simultaneously.

To address this complexity, researchers often transform MOO problems into single-objective optimization problems using a technique called scalarization. This technique involves various methods, such as the interactive weighted Tchebycheff procedure (Steuer & Choo, 1983), sequences of scalarizing functions (Luc et al., 2005), and extended forms of the generalized Tchebycheff norm (Luc et al., 2005). Additionally, scalarization techniques have been developed to generate disconnected Pareto fronts (Burachik et al., 2014). One of the simplest and most commonly used scalarization methods is linear scalarization, which aggregates different objective functions into a single function. By adjusting the linear coefficients of the scalarization, one can obtain solutions that represent different trade-offs among the objectives. However, linear scalarization has inherent limitations, as it can only identify supported solutions on the convex hull of the objective set.

Along these lines, recent work on aligning foundation models with multiple human values has led to the development of a computational method called Rewarded Soup (Rame et al., 2023). The main idea is to separately fine-tune multiple versions of a network, each optimized for a particular reward function, and then aggregate these individually fine-tuned networks by linearly interpolating their weights. Specifically, let the aligned model be represented by $p_w$, with $w$ being the neural network weights. Given reward functions $r_i$, $i = 1, \ldots, m$, let $p_{w_i}$ be the solution to the single-value alignment problem (1) with $R = r_i$. Then, rewarded soup uses the set

$$\left\{w \overset{\Delta}{=} \sum_{i=1}^{m} \rho_i w_i : \boldsymbol{\rho} \in \mathbb{S}_m\right\},$$

where $\mathbb{S}_m \overset{\Delta}{=} \{\boldsymbol{\rho} : \sum_{i=1}^{m} \rho_i = 1, \rho_i \geq 0\}$ is the $m$-dimensional simplex, to approximate the set of fine-tuned models that would result from optimizing a single reward formed by various linear combinations of the original reward functions, namely $R \overset{\Delta}{=} \sum_{i=1}^{m} \rho_i r_i$. This method aims to compute various trade-offs without the need for multiple, time-consuming training processes imposed by exploring different linear combinations.

Beyond standard preference alignment, safety-aligned models must also be resilient to adversarial threats, such as prompt injection attacks (Xian et al., 2025) and backdoor attacks (Xian et al., 2023a;b; Wang et al., 2024). A promising direction for future work is to construct preference datasets that explicitly capture these threats and develop alignment strategies that integrate both model alignment techniques and early-stage detection methods (Wang et al., 2025). Effectively combining safety detection with alignment approaches could enhance reliability and security of AI systems in real-world deployment.

### B.6 SOURCE OF REWARD FUNCTIONS

We discuss common methods for obtaining reward functions $r_i$:

**(i) From pretrained classifiers**: Each reward function associated with a preference class $v_i$ is defined as:

$$r_i(x, y) \stackrel{\Delta}{=} \log p(v_i \mid x, y) \in (-\infty, 0], \tag{24}$$

where $p(v_i \mid x, y)$ denotes the probability that a sentence $(x, y)$ belongs to the preference class $v_i$. For instance, for $v_i$ representing "harmlessness," $p(v_i \mid x, y)$ might be sourced from a classifier trained to detect if text is harmless. Assuming conditional independence of $v_i$ given $(x, y)$ and $\lambda_i = 1$ for all $i$, the solution for $q$ in (4) reduces to:

$$p(y \mid x) \propto p_0(y \mid x) \prod_{i=1}^{m} p(v_i \mid x, y) \propto p(y \mid x, v_1, \ldots, v_m). \tag{25}$$

In this configuration, the generative model $p$ functions as a "controlled generator," producing text conditioned on specified human values. For more refined applications, the model may employ $p(v_i \mid x, y)^{\lambda_i}$, allowing for the adjustment of preferences through the tempering of likelihoods. From a modeling perspective, the choice of weights $\lambda_i$ could reflect some suspicion of mis-specification of models or hyperparameters (Holmes & Walker, 2017; Ding et al., 2018; Zhang et al., 2023).

**(ii) From pairwise comparison datasets**: Rewards can also be derived from standard functions learned explicitly through pairwise or ranking comparisons, a method commonly employed in the RLHF literature.

**(iii) From language processing metrics**: Metrics such as coherence, diversity, and perplexity are used to construct rewards that reflect various aspects of language quality and effectiveness. These metrics help in refining the model to produce more desirable text characteristics.

## C PROOF OF THEOREM 1

*Proof.* For notational simplicity, we remove the conditional on $x$ and prove for the unconditional problem:

$$\min_{p \in \mathcal{P}} \mathbb{E}_{y \sim p} D_{\text{KL}}(p \,\|\, p_0), \quad \text{s.t. } \mathbb{E}_{y \sim p} r_m(y) \geq c_m, \quad m \in \{1, \cdots, M\}. \tag{26}$$

The same argument can be generalized to handle the conditional case without essential difficulty.

To align with the standard Karush-Kuhn-Tucker (KKT) conditions, where multiplies are non-negative for constraints expressed $\leq 0$, we can reformulate the original constraint as $c - \int r(y)p(y) \leq 0$. Then, we introduce Lagrange multipliers $\boldsymbol{\lambda} \geq \mathbf{0}$ for the inequality constraints and $\eta$ for the normalization constraint $\int p(y)dy = 1$. The Lagrangian for this optimization problem is:

$$L(p, \lambda, \eta) = \int p(y) \log \frac{p(y)}{p_0(y)} dy + \sum_{i=1}^{m} \lambda_i \left( c_i - \int r_i(y)p(y)dy \right) + \eta \left( \int p(y)dy - 1 \right). \tag{27}$$

To minimize the Lagrangian with respect to $q$, we take the functional derivative and set it to zero:

$$\frac{\delta L}{\delta p} = \log \frac{p(y)}{p_0(y)} + 1 - \sum_{i=1}^{m} \lambda_i r_i(y) + \eta = 0. \tag{28}$$

Rearranging terms gives:

$$p(y) = p_0(y) e^{-1 - \eta + \sum_{i=1}^{m} \lambda_i r_i(y)} = p_0(y) e^{-1 - \eta + \boldsymbol{\lambda}^{\mathrm{T}} \boldsymbol{r}(y)}. \tag{29}$$

To ensure that $p(y)$ is a probability density, it must integrate to one. Therefore,

$$\int p(y)dy = \int p_0(y) e^{-1 - \eta + \boldsymbol{\lambda}^{\mathrm{T}} \boldsymbol{r}(y)} dy = 1, \tag{30}$$

which leads to

$$e^{-1-\eta} = \frac{1}{\int p_0(y)e^{\boldsymbol{\lambda}^{\mathrm{T}}\boldsymbol{r}(y)}dy}. \tag{31}$$

So the expression for $p(y)$ becomes

$$p(y) = \frac{p(y)e^{\boldsymbol{\lambda}^{\mathrm{T}}\boldsymbol{r}(y)}}{\int p_0(y)e^{\boldsymbol{\lambda}^{\mathrm{T}}\boldsymbol{r}(y)}dy}. \tag{32}$$

Next, we prove that the above $\boldsymbol{\lambda}$ is the unique solution to (5). It can be verified that (26) is a convex minimization problem with inequality constraints. Thus, we can obtain the optimal $\lambda$ by solving the Lagrange dual problem. To that end, we derive the dual function $g(\boldsymbol{\lambda})$ by plugging the expression for $p(y)$ back into the Lagrangian (27). To highlight the dependency of $p(y)$ on $\boldsymbol{\lambda}$, we write $p_{\boldsymbol{\lambda}}(y)$. The first term in (27), the KL-divergence, becomes

$$\int p_{\boldsymbol{\lambda}}(y) \log \frac{p_{\boldsymbol{\lambda}}(y)}{p_0(y)} = \int p_{\boldsymbol{\lambda}}(y)(\boldsymbol{\lambda}^{\mathrm{T}}\boldsymbol{r}(y) - \log Z(\boldsymbol{\lambda}))dy = \int p_{\boldsymbol{\lambda}}(y)\boldsymbol{\lambda}^{\mathrm{T}}\boldsymbol{r}(y)dy - \log Z(\boldsymbol{\lambda}), \tag{33}$$

where $Z(\boldsymbol{\lambda}) = \int p_0(y)e^{\boldsymbol{\lambda}^{\mathrm{T}}\boldsymbol{r}(y)}$ is the normalizing constant which does not depend on $y$. Therefore, the Lagrangian (27) becomes

$$g(\boldsymbol{\lambda}) \triangleq \min_{p \in \mathcal{P}} L(p, \boldsymbol{\lambda}, \eta) = L(p_{\boldsymbol{\lambda}}, \boldsymbol{\lambda}, \eta)$$

$$= \int p_{\boldsymbol{\lambda}}(y)\boldsymbol{\lambda}^{\mathrm{T}}\boldsymbol{r}(y)dy - \log Z(\boldsymbol{\lambda}) + \sum_{i=1}^{m} \lambda_i \left( c_i - \int r_i(y)p_{\boldsymbol{\lambda}}(y)dy \right)$$

$$= -\log Z(\boldsymbol{\lambda}) + \boldsymbol{\lambda}^{\mathrm{T}}\boldsymbol{c}. \tag{34}$$

Finally, Lemma 1 implies that $g$ is strictly concave given that $\boldsymbol{r}(x, y)$ is not a constant. This concludes the proof of the optimal $\lambda$ being the unique solution to (5). $\qquad\square$

## D   LEMMA 1 AND ITS PROOF

**Lemma 1.** For an expectation $\mathbb{E}$ over any distribution of $(x, y)$, the Hessian of $-\log \mathbb{E}(e^{\boldsymbol{\lambda}^{\mathrm{T}}\boldsymbol{r}(x_j, y_j)}) + \boldsymbol{\lambda}^{\mathrm{T}}\boldsymbol{c}$ with respect to $\boldsymbol{\lambda}$ is non-positive definite.

*Proof.* For any probability distribution of $(x, y)$, including the empirical expectation defined by a finite sample, let $\mathbb{E}$ denotes its associated expectation. Then, it can be calculated that

$$\nabla^2_{\boldsymbol{\lambda}\boldsymbol{\lambda}}\left(-\log \mathbb{E}e^{\boldsymbol{\lambda}^{\mathrm{T}}\boldsymbol{r}(x_j, y_j)} + \boldsymbol{\lambda}^{\mathrm{T}}\boldsymbol{c}\right) \tag{35}$$

$$= \frac{(\mathbb{E}e^{\boldsymbol{\lambda}^{\mathrm{T}}\boldsymbol{r}(x,y)}\boldsymbol{r}(x,y))(\mathbb{E}e^{\boldsymbol{\lambda}^{\mathrm{T}}\boldsymbol{r}(x,y)}\boldsymbol{r}(x,y)^{\mathrm{T}}) - \mathbb{E}e^{\boldsymbol{\lambda}^{\mathrm{T}}\boldsymbol{r}(x,y)}\mathbb{E}(e^{\boldsymbol{\lambda}^{\mathrm{T}}\boldsymbol{r}(x,y)}\boldsymbol{r}(x,y)\boldsymbol{r}(x,y)^{\mathrm{T}})}{(\mathbb{E}e^{\boldsymbol{\lambda}^{\mathrm{T}}\boldsymbol{r}(x,y)})^2}.$$

Due to the Cauchy–Schwarz inequality, the above quantity is non-positive definite; Moreover, it is negative definite if and only if $\boldsymbol{r}(x, y)$ is not a constant on the support set of $x, y$. This concludes the proof of Lemma 1. $\qquad\square$

## E   PROOF OF THEOREM 2

*Proof.* For a single value alignment problem with reward function $(x, y) \mapsto r(x, y)$, the problem (1) admits a solution in the form of

$$p(y \mid x) = \frac{1}{Z(x)} \propto p_0(y \mid x)e^{\beta^{-1}R(x,y)}, \tag{36}$$

where $Z(x)$ is a normalizing that only depends on the prompt $x$. Therefore, with $R(x, y) \triangleq \beta \cdot \boldsymbol{\lambda}(\boldsymbol{c})^{\mathrm{T}}\boldsymbol{r}(x, y)$, Equations (36) and (4) are the same.

$\qquad\square$

## F    PROOF OF THEOREM 3

*Proof.* We first prove that

$$\mathcal{V}_{\text{MAP}}(r_1, \ldots, r_m; p_0) = \mathcal{V}_{\text{RLHF}}(p_0). \tag{37}$$

For every $q \in \mathcal{V}_{\text{MAP}}(r_1, \ldots, r_m; p_0)$, by definition, there exists $c$ such that $q$ is the solution to the MAP problem in (3). By invoking Theorem 1, the solution can be written as

$$p(y \mid x) = \frac{1}{Z(x, \lambda)} p_0(y \mid x) e^{\lambda^{\text{T}} r(x, y)}, \tag{38}$$

where $r \triangleq [r_1, \ldots, r_m]^{\text{T}}$, for some $\lambda = \lambda(c)$.

Meanwhile, for a single value alignment problem with reward function $(x, y) \mapsto R(x, y)$, by using calculus of variations, the formulation in (1) admits a solution in the form of

$$p(y \mid x) = \frac{1}{Z(x, \beta)} \propto p_0(y \mid x) e^{\beta^{-1} R(x, y)},$$

which includes the Equation (38). Therefore, $\mathcal{V}_{\text{MAP}}(r_1, \ldots, r_m; p_0) \subseteq \mathcal{V}_{\text{RLHF}}(p_0)$.

On the other hand, for every $q_0 \in \mathcal{V}_{\text{RLHF}}(p_0)$, it must be the solution to the following MAP problem:

$$\min_{p \in \mathcal{P}} \mathbb{E}_{x \sim \mathcal{D}, y \sim p(\cdot | x)} D_{\text{KL}}\big(p(\cdot \mid x) \,\|\, p_0(\cdot \mid x)\big)$$

$$\text{s.t. } \mathbb{E}_{x \sim \mathcal{D}, y \sim p(\cdot | x)} r_i(x, y) \geq c_i \triangleq \mathbb{E}_{x \sim \mathcal{D}, y \sim p_0} r_i(x, y), \quad \forall i = 1, \ldots, m.$$

This implies that $\mathcal{V}_{\text{RLHF}}(p_0) \subseteq \mathcal{V}_{\text{MAP}}(r_1, \ldots, r_m; p_0)$. Thus, we have proved Equality 37.

Then, with a similar argument, we can prove that

$$\mathcal{V}_{\text{MAP}}(r_1, \ldots, r_m; p_0) = \mathcal{V}_{\text{RLHF}}(r_1, \ldots, r_m; p_0). \tag{39}$$

Combining Equalities (37) and (39), we conclude the proof. □

## G    PROOF OF THEOREM 4

*Proof.* The proof of (i) and (ii) directly follows from the definition of $c$ in the MAP problem.

To prove (iii), we first calculate the gradient of $g(\lambda)$ with respect to $\lambda$:

$$\nabla_{\lambda} g(\lambda) = -\frac{\mathbb{E}_{x \sim \mathcal{D}, y \sim p_0(\cdot | x)}\big(e^{\lambda^{\text{T}} r(x, y)} r(x, y)\big)}{\mathbb{E}_{x \sim \mathcal{D}, y \sim p_0(\cdot | x)} e^{\lambda^{\text{T}} r(x, y)}} + c$$

$$= c - \mathbb{E}_{x \sim \mathcal{D}, y \sim p_{\lambda}(\cdot | x)} r(x, y),$$

where

$$p_{\lambda}(y \mid x) \triangleq \frac{p_0(y \mid x) \cdot e^{\lambda^{\text{T}} r(x, y)}}{\mathbb{E}_{x \sim \mathcal{D}, y \sim p_0(\cdot | x)} e^{\lambda^{\text{T}} r(x, y)}}.$$

As $\lambda_i \to \infty$, for any component $i$, the expected value $\mathbb{E}_{x \sim \mathcal{D}, y \sim p_{\lambda}(\cdot | x)} r(x, y)$ tends to approach the essential supreme of $r_i(x, y)$ over the support of $x \sim \mathcal{D}, y \sim p_0(\cdot \mid x)$, which is $B_i$. This implies that if $c_i > B_i$ for any $i$, the derivative $\nabla_{\lambda} g(\lambda)$ will remain positive as $\lambda_i$ increases, pushing $\lambda_i$ towards infinity without reaching a maximum for $g(\lambda)$.

To prove (iv), recall the dual function and its gradient:

$$g(\lambda) = -\log \mathbb{E}_{x \sim \mathcal{D}, y \sim p_0(\cdot | x)}\{e^{\lambda^{\text{T}} r(x, y)}\} + c^{\text{T}} \lambda,$$

$$\nabla_{\lambda} g(\lambda) = c - \mathbb{E}_{x \sim \mathcal{D}, y \sim p_{\lambda}(\cdot | x)}\{r(x, y)\}, \tag{40}$$

where the distribution $p_{\lambda}$ is defined as

$$p_{\lambda}(y \mid x) \triangleq \frac{p(y \mid x) e^{\lambda^{\text{T}} r(x, y)}}{Z(\lambda, x)}. \tag{41}$$

Let $J = \nabla^2_{\boldsymbol{\lambda}\boldsymbol{\lambda}} g(\boldsymbol{\lambda})$ be the Hessian matrix of the dual function in the space of $\boldsymbol{\lambda}$. Treating the optimal solution $\boldsymbol{\lambda} = \boldsymbol{\lambda}(\boldsymbol{c})$ as a function of $\boldsymbol{c}$, we use (40) and the implicit function theorem to obtain

$$\frac{d\boldsymbol{\lambda}(\boldsymbol{c})}{d\boldsymbol{c}} = -J^{-1}, \tag{42}$$

given that the matrix $J$ is non-singular. It remains to prove that

$$J = -Var(\boldsymbol{r}(x,y)). \tag{43}$$

Given Equation (40), it is equivalent to showing that

$$\nabla_{\boldsymbol{\lambda}} \mathbb{E}_{x\sim\mathcal{D}, y\sim p_{\boldsymbol{\lambda}}(\cdot|x)}\{\boldsymbol{r}(x,y)\} = Var(\boldsymbol{r}(x,y)). \tag{44}$$

In fact, by directly calculating the derivatives on the left-hand side, we have

$$\nabla_{\lambda_k} \mathbb{E}_{x\sim\mathcal{D}, y\sim p_{\boldsymbol{\lambda}}(\cdot|x)}\{\boldsymbol{r}(x,y)\}$$
$$= \frac{\partial}{\partial\lambda_k}\left(\frac{\mathbb{E}\{\boldsymbol{r}(x,y)e^{\boldsymbol{\lambda}^{\mathrm{T}}\boldsymbol{r}(x,y)}\}}{Z(\boldsymbol{\lambda},x)}\right)$$
$$= \frac{\mathbb{E}\{\boldsymbol{r}(x,y)r_k(x,y)e^{\boldsymbol{\lambda}^{\mathrm{T}}\boldsymbol{r}(x,y)}Z(\boldsymbol{\lambda},x)\} - \mathbb{E}\{\boldsymbol{r}(x,y)e^{\boldsymbol{\lambda}^{\mathrm{T}}\boldsymbol{r}(x,y)}\}\mathbb{E}\{r_k(x,y)e^{\boldsymbol{\lambda}^{\mathrm{T}}\boldsymbol{r}(x,y)}\}}{Z(\boldsymbol{\lambda},x)^2} \tag{45}$$

where we used $\mathbb{E}$ to abbreviate $\mathbb{E}_{x\sim\mathcal{D}, y\sim p_0(\cdot|x)}$. The quantity in (45) is the covariance of $\boldsymbol{r}(x,y)$ and $r_k(x,y)$ under $p_{\boldsymbol{\lambda}}$. We have therefore proved Identity (44) and this completes the proof.

$\square$

## H  PROOF OF THEOREM 5

*Proof.* For any $\ell = 1, 2, \ldots$, let $\ell' = (\ell \mod m)$. Because at each iteration $\ell$, we essentially aim to align $p_{(\ell)}$ such that the expected reward $r_{\ell'}$ meets a pre-defined level $c_{\ell'}$, the objective (9) becomes

$$\max_{\lambda_{\ell'}\geq 0} g_{(\ell)}(\lambda_{\ell'}) \stackrel{\Delta}{=} -\log \mathbb{E}_{x\sim\mathcal{D}, y\sim p_{(\ell-1)}(\cdot|x)}e^{\lambda_{\ell'} r_{\ell'}(x,y)} + \lambda_{\ell'} c_{(\ell')}. \tag{46}$$

Because $p_{(\ell-1)} \propto p_0 \cdot e^{\boldsymbol{\lambda}^{\mathrm{T}}_{(\ell-1)}\boldsymbol{c}}$, the problem in (46) is equivalent to

$$\max_{\lambda_{\ell'}\geq 0, \text{ Fixing } \boldsymbol{\lambda}_{\neq\ell'}=\boldsymbol{\lambda}_{(\ell-1),\neq\ell'}} g_{(\ell)}(\boldsymbol{\lambda}) \stackrel{\Delta}{=} -\log \mathbb{E}_{x\sim\mathcal{D}, y\sim p_0(\cdot|x)}e^{\boldsymbol{\lambda}^{\mathrm{T}}\boldsymbol{r}(x,y)} + \boldsymbol{\lambda}^{\mathrm{T}}\boldsymbol{c} + C_{\ell-1} \tag{47}$$

where $C_{\ell-1}$ is a term that only depends on the (fixed) $\boldsymbol{\lambda}_{(\ell)}$. As a result, solving the original objective (9) is equivalently to running coordinate-wise gradient ascent to the MAP objective in (5). This implies the convergence since the objective function in (5) is strictly concave in $\boldsymbol{\lambda}$ (Theorem 1).  $\square$

