# OpenReview forum: "MAP: Multi-Human-Value Alignment Palette"
_ICLR.cc/2025/Conference — ICLR 2025 Oral_

### Official Review · Reviewer_PidN · 2024-11-01

**Soundness:** 3
**Presentation:** 4
**Contribution:** 3
**Rating:** 8
**Confidence:** 4

**Summary:**

This paper introduces MAP, a novel approach for aligning AI with multiple human values. MAP formulates alignment as a primal-dual optimization problem with user-defined constraints, allowing precise adjustment of alignment levels based on specific user-defined values. The authors provide a robust theoretical analysis, including quantification of solution representation, sensitivity to variations in the value palette, and the feasible operational range. Extensive experiments demonstrate MAP’s effectiveness in aligning multiple values across models like OPT-1.3B and Llama2-7B-chat.

**Strengths:**

1. The theoretical foundation of MAP is robust.
2. The paper includes comprehensive experimental validation, comparing MAP with baseline methods and demonstrating its capacity to achieve desirable alignment results.
3. The paper is well-organized, with a clear presentation of the main ideas, making it easy to follow.

**Weaknesses:**

My primary concerns relate to practical implementation issues.
1. When the value palette is infeasible, the automatic adjustment process gradually reduces the target values toward the model's original performance. However, it's not clear how often infeasible palettes occur in practice and how much adjustment is typically needed. Additionally, the adjustment process requires extra calculations and iterations, which could become computationally intensive, particularly in high-dimensional, multi-value alignment scenarios. The authors should provide empirical results on these aspects and analyze and report on the computational complexity of the adjustment process for different numbers of values being aligned.
2. The decoding option is time-consuming, although it appears to offer better performance. Could you provide runtime comparisons between the decoding and finetuning options for different model sizes and numbers of values being aligned, along with corresponding performance metrics?

**Questions:**

In the decoding option, could different sample sizes $m$ affect final performance? Increasing sample size could improve performance but would also increase computational cost, suggesting a potential tradeoff. Could the authors elaborate on this?

---

> ### Author Response · Authors · 2024-11-24
> **We thank Reviewer PidN for the effort in reviewing our work and providing supportive comments. We address each concern or question below.**
>
> **Weakness 1**. When the value palette is infeasible, the automatic adjustment process gradually reduces the target values toward the model's original performance. However, it's not clear how often infeasible palettes occur in practice and how much adjustment is typically needed.  Additionally, the adjustment process requires extra calculations and iterations, which could become computationally intensive, particularly in high-dimensional, multi-value alignment scenarios. The authors should provide empirical results on these aspects and analyze and report on the computational complexity of the adjustment process for different numbers of values being aligned.
>
> **Response**. We thank the reviewer for this insightful comment. First, we acknowledge that it is not always clear in practice how often infeasible palettes occur and how much adjustment is typically needed, as this depends on two factors. The first factor is the capability of the model. Our experiments demonstrate that larger models tend to allow more room for alignment. For example, aligning the Llama2-7B-chat model with Anthropic data using the “HHH-80%” palette (enhancing Humor, Harmlessness, and Helpfulness to their respective 80th percentiles while maintaining Diversity, Coherence, and Perplexity) was feasible according to the feasibility check (Appendix A.1). However, this same palette choice was not feasible for the smaller OPT-1.3B model. The second factor is the user's palette distribution. More aggressive palettes are naturally more likely to require adjustments.
>
> Second, we emphasize that the adjustment process is computationally efficient. For both auto-adjustment strategies discussed in Appendix B.2 of the revised paper, the process relies on a one-dimensional line search irrespective of the number of values being aligned. In Appendix B.3, we provide a detailed complexity and error analysis for the primal-dual optimization problem. The computational complexity for each run of the dual optimization is $O(mn)$, where $m$ is the number of values and $n$ is the sample size. If feasibility adjustment is required, the search involves $k$ steps in a grid over (0,1], and the total complexity would scale as $O(mnk)$. In practice, a small $k$, such as 10, is typically sufficient due to the efficiency of the line search.
> To better illustrate this point, we have developed an anonymous website at https://research-demo.com/ for you to experience palette selection and auto-adjustment. We guarantee that the website does not collect any user data. Users will be able to drag sliders to enhance specific values, and the algorithm will automatically adjust the palette to ensure feasibility by interpolating between the user-specified palette and the realized value levels.
>
> To provide further empirical insights, we conducted experiments evaluating the runtime and feasibility frequency for different numbers of aligned values under two settings: Aggressive palettes (more likely to require adjustment) and Conservative palettes (inherently feasible). The results, summarized in Figure 11, Appendix B.3 of the revised manuscript, show that in the Aggressive Palette case, the feasibility frequency decreases as the number of values increases. This occurs because higher-dimensional palettes are more likely to extend beyond the Pareto frontier. Interestingly, the runtime does not increase significantly with the number of values due to the early termination of line searches when infeasible palettes are encountered. The Conservative Palette case demonstrates consistent feasibility (frequency = 1), and the runtime remains stable across different numbers of values, which shows the computational efficiency of the adjustment process.
>
> TO CONTINUE

---

> ### Author Response · Authors · 2024-11-24
> **Continued response (2/3)**
>
> **Weakness 2**. The decoding option is time-consuming, although it appears to offer better performance. Could you provide runtime comparisons between the decoding and finetuning options for different model sizes and numbers of values being aligned, along with corresponding performance metrics?
>
>
> **Response**. Thanks for the insightful comment. As suggested, we have included runtime comparisons in Appendix A.5 of the revised manuscript. Our added results highlight several key points.
>
> *Scalability with number of generations*. Firstly, for the decoding-based approach, runtime scales approximately linearly with the number of generations, as each generation requires a separate forward pass through the model. In contrast, for the PPO-based finetuning approach, the runtime does not increase significantly when scaling from $100$ to $2000$ generations. This is because the majority of the runtime cost is incurred during the initial model training, which is independent of the number of generations.
>
> *Practical choice of decoding versus fine-tuning*. Secondly, following the above point, for a small number of generations (e.g., $n=100$), decoding can be more favorable as it avoids the overhead of training. However, for a large number of generations (e.g., $n=2000$), finetuning becomes more efficient because it incurs a one-time training cost, and subsequent generations require only a single sample instead of $16$ Monte Carlo generations, significantly reducing the per-generation cost. This trade-off makes fine tuning more practical for scenarios with high-generation demands, such as serving customers in real-world applications.
>
> *Impact of model*. Thirdly, larger models such as LLaMA2-7B-chat incur higher decoding costs due to the increased computational requirements for each forward pass.
>
> *Impact of values choices*. Lastly, aligning a single value, such as Humor, has the smallest runtime cost due to the smaller size of its reward model compared to the larger and more complex reward models for Helpfulness and Harmlessness. Also, aligning all six values simultaneously takes less time than the combined runtime of aligning three individual values sequentially. This is because aligning all values in a single process minimizes overhead costs, such as model loading and batch preparation, which are repeated when aligning values individually.
>
>
> | Values to Align  	| $n$	| Llama2-7B-chat (Decoding) | OPT-1.3B (Decoding) | OPT-1.3B (Fine-tuning) |
> |----------------------|-------|---------------------------|----------------------|-------------------------|
> | **All 6 values** 	| 100   | 7.95                  	| 4.25            	| 27.16              	|
> |                  	| 2000  | 160.21                	| 107.32          	| 29.54              	|
> | **Humor only**   	| 100   | 4.45                  	| 2.32            	| 27.91              	|
> |                  	| 2000  | 90.90                 	| 37.87           	| 28.50              	|
> | **Helpfulness only** | 100   | 4.95                  	| 2.66            	| 29.45              	|
> |                  	| 2000  | 106.92                	| 50.60           	| 31.52              	|
> | **Harmlessness only**| 100   | 5.61                  	| 2.69            	| 31.23              	|
> |                  	| 2000  | 107.33                	| 51.64           	| 36.67              	|
>
> TO CONTINUE

---

> > ### Author Response · Authors · 2024-11-24
> > **Continued response (3/3)**
> >
> > **Question**. In the decoding option, could different sample sizes $m$ affect final performance? Increasing sample size could improve performance but would also increase computational cost, suggesting a potential tradeoff. Could the authors elaborate on this?
> >
> > **Response**. We thank the reviewer for this insightful question. Indeed, the sample size $k$ in the decoding option affects both the computational cost and the performance of the aligned model. Increasing $k$ can improve the approximation of the desired distribution and enhance the realized values,  but at a cost of increased computation and latency.
> > To explore this tradeoff, we added an experiment detailed in Appendix A.4 of the revised manuscript, comparing realized value levels and runtime (in minutes) for decoding-based alignment with varying Monte Carlo samples ($k$). The experiments, conducted for 1000 generations using Llama2-7B-chat under the same settings as Appendix A.1 on a single A100 GPU, average results over 3 repetitions, with a standard error of realized levels within 0.02 and runtime within 0.5 minutes. For better readability, we subtracted the numbers for each $k$ by those at $k=16$ as a reference. The results demonstrate that increasing $k$ significantly improves performance up to $k=8$, with diminishing returns beyond this point. Notably, $k=8$ achieves realized values nearly on par with $k=16$.
> > | $k$ | Runtime | Humor  | Helpful | Harmless | Diversity | Coherence | Perplexity |
> > |-----|---------|--------|---------|----------|-----------|-----------|------------|
> > | 2   | 12.90   | -0.15  | -0.09   | -0.42	| -0.00 	| -0.02 	| -0.04  	|
> > | 4   | 15.87   | -0.05  | -0.02   | -0.28	| -0.00 	| 0.01  	| 0.01   	|
> > | 8   | 20.70   | -0.01  | -0.05   | -0.05	| -0.01 	| -0.01 	| 0.00   	|
> > | 16  | 32.07   | 0.00   | 0.00	| 0.00 	| 0.00  	| 0.00  	| 0.00   	|

---

> > > ### Comment · Reviewer_PidN · 2024-11-25
> > >
> > > Thank you for the detailed response and for providing the demo. This has addressed my concerns, and I will raise my score accordingly.

---

> > > > ### Author Response · Authors · 2024-11-25
> > > > **Thank you for raising the score**
> > > >
> > > > Dear Reviewer PidN,
> > > >
> > > > We sincerely appreciate the time and effort you dedicated to reviewing our paper. Your invaluable comments and insights have been extremely helpful in improving our work. We are also grateful for your decision to raise the score.
> > > >
> > > > Best regards,
> > > >
> > > > Authors

---

### Official Review · Reviewer_AExF · 2024-11-06

**Soundness:** 3
**Presentation:** 4
**Contribution:** 3
**Rating:** 8
**Confidence:** 4

**Summary:**

The paper introduces MAP, a novel approach for aligning generative AI systems with multiple human values, addressing the complexity of value trade-offs through an optimization framework with user-defined constraints. It provides a theoretical analysis of value trade-offs, sensitivity to constraints, and establishes the sufficiency of linear weighted rewards for multi-value alignment, supported by extensive empirical studies demonstrating MAP's effectiveness across various tasks. The paper contributes to the field by offering a structured method for value alignment that holds potential for positively impacting complex decision-making domains, with future work suggested to extend MAP's application to empirical risk calculations using mixed data sources.


However, to further strengthen the paper, the authors could provide more insights into practical challenges such as scalability, robustness to reward model uncertainties, and handling of non-linear value interactions. Additionally, a deeper exploration of ethical considerations and user guidance for feasible value palette specification would enhance the impact and applicability of the approach in real-world settings.

**Strengths:**

s1
The paper introduces a novel and principled formulation of the multi-human-value alignment problem. By introducing user-defined value palettes and framing alignment as a constrained optimization task, they provide a flexible and interpretable method for aligning AI models with complex human value systems. Meanwhile, the problem setup, i.e. a “Pallette”, also allows a user-friendly framework for future adaptation in real case alignment applications.

s2
The paper makes rigorous theoretical analysis of the problem and corresponding solutions,  including proofs of key properties such as the equivalence of realizable value levels between MAP and traditional RLHF methods. It demonstrates that linear combinations of individual reward functions are sufficient to capture the entire Pareto frontier in the value space(like shown in Theorem 3), which is a significant insight for multi-value alignment.

s3
The introduction of primal-dual method as a solution is both natural and practical.  The algorithm includes feasibility checks for user-specified value palettes and provides guidance on adjusting infeasible palettes, making it applicable in real-world scenarios where computational resources and feasibility constraints are critical.

**Weaknesses:**

w1
While the paper discusses the efficiency of the primal-dual approach, it does not thoroughly address how the computational complexity scales with larger models (e.g., models with hundreds of billions of parameters) or with an increasing number of value dimensions. Practical implementations on very large-scale models may face computational challenges.

Generally the primal-dual method requires the computation of gradients with respect to the dual variables (λ), which involves expectations over the model's outputs. When dealing with large models or a large number of human values (i.e., higher-dimensional λ), the computational burden increases.

Therefore it will be helpful if the paper introduces more experimental study with larger models and settings with more values and discusses the performance of the methods when it scales.

w2
The primal-dual method relies on accurate reward functions (r_i) to guide the optimization. If these reward functions are noisy, biased, or mis-specified, the dual variables λ may be estimated incorrectly. This can lead to suboptimal or unintended alignments.

Also, computationlly, the primal-dual solution settings may have certain infeasible inputs(like stated in section 2.3 as well), this might further limit the application of complicated value alignment cases.

w3
This is related to w2. Generally, the concavity of the dual function is crucial for guaranteeing convergence to a global maximum using gradient ascent. However, in practice, especially when dealing with large scale foundation models, which may present non-convex loss landscapes, the assumptions may not hold strictly.

Also as described in equations(7) and equations(8), the expectations are approximated with finite samples, which may introduce not only errors but also non-convexities. And this may lead to unconvergence of the primal-dual method.

It will be helpful if the paper could also provide discussion and empirical studies over these concerns. Also, though might be very hard, it will be great if the paper could provide a theoretical discussion concerning the convergence of the primal-dual solution in the paper’s setting, or empirically show if off-the-shelf models generates convergence problem(like saddle points) and how to deal with them.

**Questions:**

As most technical concerns are addressed in strong points and weak points, here are some questions and further comments:
* How does MAP perform when reward models are uncertain or biased? Are there mechanisms within MAP to mitigate negative effects from mis-specified rewards or to assess the robustness of the alignment?
* Like stated in section 2.3, "MAP either prompts the user to adjust the value palette or automatically modifies 𝑐 c to ensure feasibility.”, is there any quantitative/qualitative guidance for users to adjust their value to enhance usability? If yes, what are the technical framework?
* To deal with the computational overhead and scalability problem, is it feasible to further adapt the primal-dual method with a more stochastic manner or make it a bi-level optimization primal-dual setup?

---

> ### Author Response · Authors · 2024-11-24
> **We thank Reviewer AExF for the effort in reviewing our work and providing supportive comments. We address each concern or question below.**
>
> **Weakness 1**. While the paper discusses the efficiency of the primal-dual approach, it does not thoroughly address how the computational complexity scales with larger models (e.g., models with hundreds of billions of parameters) or with an increasing number of value dimensions. Practical implementations on very large-scale models may face computational challenges. Generally the primal-dual method requires the computation of gradients with respect to the dual variables ($\boldsymbol{\lambda}$), which involves expectations over the model's outputs. When dealing with large models or a large number of human values (i.e., higher-dimensional $\boldsymbol{\lambda}$), the computational burden increases. Therefore it will be helpful if the paper introduces more experimental study with larger models and settings with more values and discusses the performance of the methods when it scales.
>
> **Response**. We thank the reviewer for raising concerns about the computational complexity of the MAP optimization process. Below, we summarize our analysis, which demonstrates that the approach is computationally efficient and scales well for practical implementations.
> The optimization involves calculating the dual objective (7) in the paper that approximates the original objective (5) involving expectation. The objective involves $n$ as the number of Monte Carlo (MC) samples and $m$ as the number of reward dimensions. The computational complexity per iteration is $O(mn)$, and the memory complexity is also $O(mn)$. We have included a detailed analysis in Appendix B.3 of the revised paper. Importantly, this complexity does not depend on the size of the model being aligned, as the MC samples are generated from the pre-trained model before optimization and can be reused for various palette choices.
> A natural question to ask then is how to choose the number of MC samples  $n$ to approximate the expectation. To quantify how $n$ affects the accuracy of the dual solution, we analyzed the deviation between the solution $\boldsymbol{\lambda}_n$ obtained with $n$ samples and the ideal solution $\boldsymbol{\lambda}$ with infinite samples. Because the dual objective function under both expectation and finite-sample average are concave, under reasonable assumptions, the required $n$ grows quadratically with the desired precision $\epsilon^{-2}$. In practice, we found that thousands of MC samples suffice to achieve accurate solutions. We have also added Figure 12 in Section B.3 of the revised manuscript to visualize the concavity of the dual objective function for MAP alignment.
>
> **Weakness 2**. The primal-dual method relies on accurate reward functions (r_i) to guide the optimization. If these reward functions are noisy, biased, or mis-specified, the dual variables $\boldsymbol{\lambda}$ may be estimated incorrectly. This can lead to suboptimal or unintended alignments. Also, computationally, the primal-dual solution settings may have certain infeasible inputs(like stated in Section 2.3 as well), this might further limit the application of complicated value alignment cases.
>
> **Response**. We thank the reviewer for the insightful comments. To address the concern regarding noisy or mis-specified reward functions, we have added Appendix B.1 titled “Perturbation Analysis of MAP under Mis-specified Rewards” in the revised manuscript. There, we analyze the sensitivity of the dual variable $\boldsymbol{\lambda}$ and the aligned distribution $p$ to perturbations in the rewards $\mathbf{r}$.
>
> Specifically, if the operational reward $\mathbf{r}$ deviates from the true reward $\mathbf{r}_0$ by at most $\delta$ in sup-norm, we derive the following bounds in Appendix B.1.1:
>
> $\|\boldsymbol{\lambda} - \boldsymbol{\lambda}_0\|_2 $
>
> $\leq \frac{\delta}{\lambda_{\min}}$,
>
> $D_{\mathrm{KL}}(p_{\boldsymbol{\lambda}} \| p_{\boldsymbol{\lambda}_0})$
>
> $\leq \frac{2\delta (B_0 + \delta)}{\lambda_{\min}}.$
>
> Here, $\lambda_{\min}$ is the smallest eigenvalue of the covariance matrix of rewards under $p$, and $B_0$ is the maximum magnitude of the true rewards $\mathbf{r}_0$.
>
> These results demonstrate that the MAP framework remains robust to small perturbations in the reward functions, especially when the rewards are diverse and $\lambda_{\min}$ is sufficiently large.
>
> Additionally, we analyze continuous perturbations by modeling the reward deviation as
>
> $\mathbf{r}_t = \mathbf{r}_0 + t (\mathbf{r} - \mathbf{r}_0)$,
>
> where $t \in [0, 1]$ in Appendix B.1.2. Our result provides insights into how $\boldsymbol{\lambda}$ evolves with perturbations, with the deviation represented as an integral of infinitesimal changes. These analyses confirm the robustness of MAP under noisy or mis-specified rewards.
>
> TO CONTINUE

---

> ### Author Response · Authors · 2024-11-24
> **Continued response (2/3)**
>
> To address the concern regarding infeasibility, we emphasize that detecting and addressing infeasible palettes is an integral part of our approach. In Appendix B.2, we proposed two automatic strategies to adjust an infeasible palette $\mathbf{c}$. The first strategy iteratively interpolates between an infeasible $\mathbf{c}$ and the realized value level $\mathbf{c}_0$ under the original model $p_0$. By gradually moving $\mathbf{c}$ toward $\mathbf{c}_0$, a feasible solution is identified while staying as close as possible to the user's specified palette. The second strategy is to prioritize specific values. For example, we increase $c_1$ to its maximum feasible level while ensuring the realized levels of other values remain unchanged. This process can then be repeated for other values in a user-specified priority order. Both strategies are efficient since they use line search. For clarity, we have included pseudocode for these methods in Appendix B.2 and added a short experimental study in Figure 11, Section B.3 of the revised manuscript.
>
> **Weakness 3**. This is related to Weakness 2. Generally, the concavity of the dual function is crucial for guaranteeing convergence to a global maximum using gradient ascent. However, in practice, especially when dealing with large scale foundation models, which may present non-convex loss landscapes, the assumptions may not hold strictly.
>
> Also as described in equations(7) and equations(8), the expectations are approximated with finite samples, which may introduce not only errors but also non-convexities. And this may lead to unconvergence of the primal-dual method.
>
> It will be helpful if the paper could also provide discussion and empirical studies over these concerns. Also, though might be very hard, it will be great if the paper could provide a theoretical discussion concerning the convergence of the primal-dual solution in the paper’s setting, or empirically show if off-the-shelf models generates convergence problem(like saddle points) and how to deal with them.
>
> **Response**. We thank the reviewer for raising these thoughtful concerns. We address them through the following points.
>
> First, we would like to point out that the convergence of the primal-dual optimization does not depend on the size of the model being aligned. As described in the revised paper (Section 2.3 and Lemma 1 in Appendix D), the dual objective function is always concave with respect to $\boldsymbol{\lambda}$ under our problem formulation, whether it be involving true expectation or finite samples approximation (which is essentially an expectation operator over a finite measure). This provides a theoretical guarantee of the convergence of the primal-dual method **irrespective of the non-convexity of the underlying model being aligned**. We have also added Figure 12 in Appendix B.3 of the revised manuscript.
>
> Following your comment, we have added Appendix B.3 in the revised manuscript to study the error of using Monte Carlo samples to approximate expectations in the dual objective. We show that the deviation between the finite-sample solution $\boldsymbol{\lambda}_n$ and the true solution $\boldsymbol{\lambda}$ (with infinite samples) is bounded as:
>
> $ \|\boldsymbol{\lambda}_n - \boldsymbol{\lambda}\|_2$
>
> $\leq \frac{C}{\lambda_{\min} \sqrt{n}},$
>
> where $C$ is a constant depending on the boundedness of rewards and $\lambda_{\min}$ is the smallest eigenvalue of the reward covariance matrix. This result demonstrates that to achieve a deviation
>
> $\|\boldsymbol{\lambda}_n - \boldsymbol{\lambda}\|$
>
> $\leq \epsilon$,
>
> the required number of MC samples scales as $n = \Omega(\lambda_{\min}^{-2} \epsilon^{-2})$. We hope this theoretical analysis can  address concerns about finite-sample errors and ensure that the method is reliable for practical applications.
>
> **Question 1**. How does MAP perform when reward models are uncertain or biased? Are there mechanisms within MAP to mitigate negative effects from mis-specified rewards or to assess the robustness of the alignment?
>
> **Response**. We thank the reviewer for this insightful question. As discussed in response to Weakness 2 and detailed in Appendix B.1, we have conducted a comprehensive sensitivity analysis of the MAP framework to mis-specified rewards. These results demonstrate that MAP remains robust under small reward perturbations and provide theoretical bounds on the deviation in the aligned distribution and dual variables.
>
> TO CONTINUE

---

> ### Author Response · Authors · 2024-11-24
> **Continued response (3/3)**
>
> **Question 2**. Like stated in section 2.3, "MAP either prompts the user to adjust the value palette or automatically modifies 𝑐 c to ensure feasibility.”, is there any quantitative/qualitative guidance for users to adjust their value to enhance usability? If yes, what is the technical framework?
>
> **Response**. We thank the reviewer for this question. As discussed in our response to Weakness 2 and detailed in Section 2.3 and Appendix B.2, MAP provides two automatic adjustment strategies for infeasible palettes: (1) interpolating between the infeasible palette and the realized value level to ensure feasibility, and (2) prioritizing specific values by maximally enhancing one while maintaining others, both via line search. To further illustrate this, we have worked over the past few days to develop an anonymous website featuring a **simple and intuitive UI/UX** for you to experience palette selection and auto-adjustment. We guarantee that the website does not collect any user data. The URL for the demo is https://research-demo.com/.
>
> **Question 3**. To deal with the computational overhead and scalability problem, is it feasible to further adapt the primal-dual method with a more stochastic manner or make it a bi-level optimization primal-dual setup?
>
> **Response**. We thank the reviewer for this thoughtful question. In our current work, during the pre-alignment phase, the reward computation for Monte Carlo samples from the original model can be fully parallelized. This ensures that all reward models do not need to be loaded simultaneously into memory to reduce computational and memory overhead.
>
> When aligning the model using the MAP-optimized $\boldsymbol{\lambda}$, we acknowledge that loading all reward models simultaneously could be memory-intensive. To address this, Section 2.4 introduces a sequential alignment strategy where individual or subsets of values are aligned iteratively. Theorem 2 in the paper demonstrates that this sequential approach, which operates as a stochastic-like gradient ascent, guarantees convergence to the MAP solution over multiple cycles. This strategy provides an effective way to handle constraints such as limited GPU memory or computational resources.
>
> We also appreciate the reviewer's suggestion regarding potential extensions to a bi-level optimization framework. Incorporating adversarial agents that perturb the rewards or the model itself could provide a valuable robustness mechanism, enhancing the practical applicability of MAP. While this is beyond the current scope, we consider it a fantastic direction for future work.

---

> ### Comment · Reviewer_AExF · 2024-11-26
>
> Thanks for the feedback and supplementary analysis, which have much addressed my concerns in reviewing the paper. While not all analysis and improvements could be incorporated into the current draft, I am confident that they will pave the way for productive future work.
>
> The rating will be updated accordingly.

---

> > ### Author Response · Authors · 2024-11-26
> > **Thank you for raising the score**
> >
> > Dear Reviewer AExF,
> >
> > We sincerely thank you again for reviewing our paper and providing instrumental comments for improving it, and we truly appreciate your decision to increase the score.
> >
> > Best regards,
> > Authors

---

### Official Review · Reviewer_so1s · 2024-11-07

**Soundness:** 3
**Presentation:** 4
**Contribution:** 3
**Rating:** 8
**Confidence:** 4

**Summary:**

This work studies the problem of aligning multiple human values in generative AI systems. The key challenge is the potential conflicts between human values, which require the model to adapt in varying directions. To address this, the authors propose to first define each value principle as a constraint in model optimization process. They demonstrate, through a primal-dual approach, that this constrained optimization problem is equivalent to optimizing a linear combination of multiple values. Their analysis shows the proposed method could reach the Pareto Frontier. The optimization problem is solved via gradient ascent approach. An extensive evaluation on two datasets with multiple value dimensions show that the proposed method could adjust the model to reflect diverse human preferences.

**Strengths:**

The proposed constrained formulation to address multiple value alignment problem introduces a novel perspective, and their primal-dual analysis shows an interesting mapping between such formulation and a linear weighted combination formulation. Both theoretical analysis and experiment evaluations are comprehensive, showing the generality and applicability of the proposed approach in realistic cases. Overall, this paper is of high quality with clear presentation.

**Weaknesses:**

One minor issue is about the interpretability of the approach. Despite some discussions on interpretations of $\lambda$ and value palette c. their practical implications and selection criteria are unclear. Another limitation is the reliance on a numerical representation for each human value, either obtained from pre-trained models or from human evaluations. However, such dependency is a constraint shared by many existing work, and falls beyond the scope of this work.

**Questions:**

1. Regarding Remark 2, could the authors clarify how to select the value palette c? For a classifier-based approach, how to set the log probability, if the objective is to make the model align with value A 20% more often than before tunning?
2. Following Question 1, if there are 3 values {A, B, C}, how to set the constraint c, if  the goal is to achieve an order A>B>C? where the model should prioritize value A alignment, follow B if A is satisfied, and align with C is both A and B are respected?
3. In the simulation experiments, since the rewards for values come from pre-trained classifiers in the form of log probabilities, would an additional step for bounding or normalization be necessary?
4. In section 3.3, $\lambda$ is sampled randomly instead of c, despite $\lambda$ being uniquely dependent on c. Could the authors clarify the reasoning behind this choice?

**Details Of Ethics Concerns:**

Not applied.

---

> ### Author Response · Authors · 2024-11-24
> **We thank Reviewer so1s for the effort in reviewing our work and providing supportive comments. We address each concern or question below.**
>
> **Question 1**. Regarding Remark 2, could the authors clarify how to select the value palette c? For a classifier-based approach, how to set the log probability, if the objective is to make the model align with value A 20% more often than before tunning?
>
> **Response**. Thanks for your comment. We have rewritten Remark 2 and added Appendices B.2&B.4 in the revised manuscript to clarify details. For your convenience, we briefly summarized it here.
>
> We have discussed three ways to select the value palette $\mathbf{c}$ in practice. The first method, which we used in the experiments (e.g., Table 1, Section 3.2) is based on the quantile of reward distributions under the original model. Users can set each component of $\mathbf{c}$ to correspond to a specific quantile. For example, for the first value, setting $c_1$ to the 80% quantile positions the realized level of the aligned model within the upper 20% of expected outcomes. As discussed in Section 2.3 and Appendix B.2, in practice, we use a pilot dataset generated from the original model as Monte Carlo samples for empirical estimation of quantiles. This pilot dataset can serve various palette configurations.
>
> The second method is the classifier-based approach, which leverages a pretrained classifier. We used this in the experiment for Table 2, Section 3.2. Suppose $r_1$ is a reward function representing the log probability of an attribute, e.g., $r_1 = \log p(\textrm{harmlessness} \mid x,y)$. If a user wants to make harmlessness 20% more likely under the aligned model $p$, this corresponds to setting the following palette:
>
> $\mathbb{E}_{x \sim \mathcal{D} , y \sim p(\cdot \mid x)} r_1(x, y)  \geq $
>
> $\mathbb{E}_{x \sim \mathcal{D} , y \sim p_0(\cdot \mid x) } r_1(x,y) + \log (1+\kappa),$
>
> where $\kappa = 0.2$ reflects the desired 20% improvement. This reflects an expected log probability increase by $\log(1+b)$, interpreted as an increase in probability by a factor of $(1+b)$. Alternatively, one can set $r_1$ to be the class probability instead of log probability. In doing this, the user can set palettes for other values to maintain their levels.
>
> The third approach is to automatically adjust the palette to ensure feasibility based on the user’s specified palette. Since not all user-specified $\mathbf{c}$ result in a feasible MAP problem, Appendix B.2 introduces two auto-adjustment strategies. One strategy is to iteratively interpolate between an infeasible $\mathbf{c}$ and the realized value level $\mathbf{c}_n$ under the original model $p_0$ until a feasible solution is achieved. Alternatively, users can prioritize specific values by increasing $c_1$ to the maximal that admits a feasible solution, while maintaining the levels of other values. In the revised manuscript, we have added two pseudocode algorithms in Appendix B.2 for clarity, and added an experimental study in Figure 11, Appendix B.3 to study the computational complexity of auto-adjustment.
>
> **Question 2**. Following Question 1, if there are 3 values {A, B, C}, how to set the constraint c, if the goal is to achieve an order A>B>C? where the model should prioritize value A alignment, follow B if A is satisfied, and align with C is both A and B are respected?
>
> **Response**. If the goal is to prioritize alignment among three values with the ordered preference: A>B>C, MAP can prioritize a particular value and set its palette, say $c_1$, to be the maximal that admits a feasible solution. That is, MAP can search for the largest $\rho \geq 0$ that $\mathbf{c}' = \mathbf{c} + [\rho, 0, \ldots,0]$ is feasible. Once A is aligned to its maximal feasible level, MAP can apply the same procedure to further enhance B while maintaining the other values (including the adjusted A) to their current levels. For clarity, we have added pseudocode for this approach in Appendix B.2, Algorithm 3.
> To better address this point, we have worked over the past few days to develop an anonymous website featuring a simple and intuitive UI/UX for you to experience palette selection and auto-adjustment firsthand. We guarantee that the website does not collect any user data. The URL for the demo is https://research-demo.com/.
>
> TO CONTINUE

---

> ### Author Response · Authors · 2024-11-24
> **Continued response (2/2)**
>
> **Question 3**. In the simulation experiments, since the rewards for values come from pre-trained classifiers in the form of log probabilities, would an additional step for bounding or normalization be necessary?
>
> **Response**. Thank you for this insightful comment. In our simulation experiments, we did not apply any bounding or normalization. This decision was based on the observation that setting the palette using quantiles provides robustness to extreme values, as the selected quantiles are not near the edges (i.e., 0 or 1). To ensure the uncertainty of the average reward is small and the conclusions are meaningful, we calculated the standard error of the mean and reported them (e.g., the shaded bands in Figure 7), which are negligible compared to the observed differences in average rewards.
> We acknowledge that log probabilities, ranging from negative infinity to zero, could lead to large variations in average rewards. While this was not an issue in our experiments, we agree that clipping extreme values could enhance stability in some cases. In the open-source package we will release, we plan to include a function that clips rewards to a small quantile (e.g., 1%) to allow users greater flexibility and control over stability.
>
>
> **Question 4**. In section 3.3, $\boldsymbol{\lambda}$ is sampled randomly instead of $\mathbf{c}$, despite $\boldsymbol{\lambda}$ being uniquely dependent on $\mathbf{c}$. Could the authors clarify the reasoning behind this choice?
>
> **Response**. Thank you for this insightful question. The reason for randomly sampling $\boldsymbol{\lambda}$ instead of $\mathbf{c}$ in Section 3.3 is that our primary goal is to compare MAP's navigation efficiency (the frequency of generating outcomes in the desirable regime) with existing baseline methods. In the literature, it is standard practice to sample $\boldsymbol{\lambda}$ randomly when evaluating methods designed to handle trade-offs among multiple objectives. Unlike MAP, which derives $\boldsymbol{\lambda}$ uniquely from a given $\mathbf{c}$, baseline methods such as MORL lack this structured perspective and instead rely on directly sampling $\boldsymbol{\lambda}$. By adopting this same setting for MAP, we demonstrate in Figure 7 that MAP achieves significantly better navigation efficiency than random sampling of $\boldsymbol{\lambda}$.

---

> ### Comment · Reviewer_so1s · 2024-11-24
>
> Thanks for the updates and the demo. After reviewing the response, I have decided to maintain my current evaluation.

---

> > ### Author Response · Authors · 2024-11-25
> > **Thank you again**
> >
> > Dear Reviewer so1s,
> >
> > We would like to thank you again for reviewing our paper and providing invaluable comments. They have been extremely helpful in improving our work.
> >
> > Best regards,
> >
> > Authors

---

### Author Response · Authors · 2024-11-24
**Overall Response to Reviewers**

We sincerely thank all reviewers for their thoughtful comments and constructive feedback, which have significantly improved our work.
In the revised manuscript, we have comprehensively addressed each concern, incorporating additional experiments, theoretical analyses, and detailed discussions. Major changes and updates are highlighted in blue for your convenience.

We have also developed a demo webpage (https://research-demo.com/) to provide an interactive experience with palette selection and automatic adjustments. We assure you that the website is entirely anonymous and does not collect any user data.

Please do not hesitate to let us know if there are any additional concerns or suggestions. We are happy to provide further clarifications or make further revisions if needed.

Thank you once again for your valuable time and insights.

---

### Meta-Review · Area_Chair_Ak5x · 2024-12-09

**Metareview:**

The paper introduces Multi-Human-Value Alignment Palette (MAP), an approach for aligning generative models with multiple human values by framing value alignment as a constrained optimization problem. The proposed method uses user-defined constraints to set alignment targets and solves the optimization problem through a primal-dual approach. The authors provide theoretical analyses, including trade-off quantification, sensitivity to constraints, and the connection between multi-value and sequential alignment. Empirical results across various datasets demonstrate MAP’s capability to align multiple values while balancing trade-offs efficiently.

Overall, the reviewers recognize the paper’s novel formulation, rigorous theoretical analysis, and comprehensive experimental evaluation. While some concerns are raised regarding scalability, computational efficiency in large models, sensitivity to reward mis-specification, and potential challenges with non-convex optimization landscapes, these are not considered to overshadow the paper's contributions. All reviewers agree that the paper should be accepted.

**Additional Comments On Reviewer Discussion:**

There is a consensus among reviewers to accept the paper.

---

### Decision · Program_Chairs · 2025-01-22

Accept (Oral)